# Epigenetic control of melanoma cell invasiveness by the stem cell factor SALL4

Johanna Diener [1,10], Arianna Baggiolini [1,2,10], Mattias Pernebrink [3,4], Damian Dalcher[5], Luigi Lerra[5], Phil F. Cheng [6], Sandra Varum[1], Jessica Häusel[1], Salome Stierli [1], Mathias Treier [7,8], Lorenz Studer [2], Konrad Basler[9], Mitchell P. Levesque[6], Reinhard Dummer [6], Raffaella Santoro [5], Claudio Cantù [3,4,9] & Lukas Sommer [1✉]

Melanoma cells rely on developmental programs during tumor initiation and progression. Here we show that the embryonic stem cell (ESC) factor Sall4 is re-expressed in the $Tyr::Nras^{Q61K}$; $Cdkn2a^{-/-}$ melanoma model and that its expression is necessary for primary melanoma formation. Surprisingly, while Sall4 loss prevents tumor formation, it promotes micrometastases to distant organs in this melanoma-prone mouse model. Transcriptional profiling and in vitro assays using human melanoma cells demonstrate that SALL4 loss induces a phenotype switch and the acquisition of an invasive phenotype. We show that SALL4 negatively regulates invasiveness through interaction with the histone deacetylase (HDAC) 2 and direct co-binding to a set of invasiveness genes. Consequently, SALL4 knock down, as well as HDAC inhibition, promote the expression of an invasive signature, while inhibition of histone acetylation partially reverts the invasiveness program induced by SALL4 loss. Thus, SALL4 appears to regulate phenotype switching in melanoma through an HDAC2-mediated mechanism.

[1] University of Zürich, Institute of Anatomy, Zürich, Switzerland. [2] Developmental Biology, The Center for Stem Cell Biology, Memorial Sloan Kettering Cancer Center, New York, NY, USA. [3] Wallenberg Centre for Molecular Medicine, Linköping University, Linköping, Sweden. [4] Department of Biomedical and Clinical Sciences, Division of Molecular Medicine and Virology; Faculty of Medicine and Health Sciences, Linköping University, Linköping, Sweden. [5] University of Zürich, Department of Molecular Mechanisms of Disease, Zürich, Switzerland. [6] University Hospital of Zürich, Department of Dermatology, Zürich, Switzerland. [7] Max-Delbrück-Center for Molecular Medicine, Berlin, Germany. [8] Charité-Universitätsmedizin Berlin, Berlin, Germany. [9] University of Zürich, Institute of Molecular Life Sciences, Zürich, Switzerland. [10] These authors contributed equally: Johanna Diener, Arianna Baggiolini. ✉email: lukas.sommer@anatomy.uzh.ch

General hallmarks of cancer include genetic mutations and chromosomal rearrangements[1]. To sustain growth and eventually progress to metastatic disease, tumors undergo several additional molecular changes including epigenetic rewiring and altered transcription of specific genes[2,3]. The aberrant re-expression of genes reminiscent of the embryonic cell-of-origin and the hijacking of its transcriptional programs have been identified as possible drivers of cancer progression[4–6]. In different cancer types, stem-cell-like cancer cells have been associated with tumor initiation, sustained growth, and metastasis formation[7]. Furthermore, the reacquisition of features reminiscent of stem cells has been associated with reduced anticancer immunity, resistance to different therapies and disease relapse[8–10].

Cutaneous melanoma is the most aggressive skin cancer due to its high metastatic potential[11]. The embryonic cell-of origin of melanocytes, from which melanoma arises, is the neural crest stem cell (NCSC)[12]. Melanoma cells are known to hijack neural crest (NC)-related migratory programs during tumor initiation, invasion, and metastasis formation[13–18]. For instance, the transcription factors PAX3, FOXD3, and SOX10 are part of a gene regulatory network in early NC development that also supports melanoma cell growth, migration, and resistance to targeted therapy, respectively. Furthermore, the transcription factor YY1 has recently been shown to control comparable metabolic pathways in NC and melanoma cells and to be equally required for both development and melanoma formation[19]. Likewise, the neurotrophin receptor CD271/NGFR/p75[NTR], which marks migratory NCSCs, is re-expressed in melanoma cells and renders them more invasive, metastatic, and therapy resistant[13,15,20,21] and of note, transient ectopic expression of NGFR was shown to promote phenotype switching—the dynamic transition of melanoma cells from a proliferative to a highly invasive state[22–24].

Here we identify the stem cell factor SALL4 as a regulator of melanoma phenotype switching. We find that Sall4 is the strongest upregulated transcription factor in hyperplastic, melanoma-prone murine melanocytes when compared to normal wild-type melanocytes and that Sall4 is crucial for primary melanoma growth. SALL4 is a known embryonic stem cell (ESC) regulator[25] and its aberrant re-expression has been reported for an increasing number of cancer types, such as germ cell tumors, hepatocellular carcinoma, gastric cancer, leukemia-, and others[26]. Due to its re-expression in cancers, SALL4 has been dubbed an 'oncofetal' gene[27]. Since such genes are not expressed in adult tissues except for malignant lesions, factors such as SALL4 represent ideal targets for cancer diagnosis and disease treatment[27,28]. However, in the present study, we show that while upregulation of Sall4 is crucial to sustain melanoma tumor growth, its depletion or downregulation increases invasiveness and metastasis formation in melanoma. Intriguingly, SALL4 appears to regulate a melanoma-specific invasion program through HDAC2-mediated epigenetic silencing of invasiveness genes.

## Results

**Sall4 is re-expressed in hyperplastic murine melanocytes**. To identify factors essentially involved in melanoma tumorigenesis, we analyzed RNA sequencing data previously obtained[19] from wild-type melanocytes and hyperplastic melanocytes isolated from the skin of 3-months-old $Tyr::Nras^{Q61K}$; $Cdkn2a^{-/-}$ mice (Fig. 1a). Loss of the tumor suppressor Cdkn2a together with gain of Nras function in the melanocytic lineage of these mice leads to hyperplastic melanocytes already at birth and primary melanoma arising around the age of 6 months[29]. When comparing hyperplastic to wild-type melanocytes by RNA sequencing (RNA seq) (Supplementary Data 1), we found among the top 20 upregulated genes one transcription factor, Sall4 (logFC 9.59, $p$ value < 0.0001)

(Fig. 1b). Upregulation of Sall4 in hyperplastic versus wild-type melanocytes was confirmed by immunohistochemistry on mouse skin sections (Fig. 1c). Moreover, murine primary melanoma displayed prominent Sall4 expression (Supplementary Fig. 1). Likewise, we could not detect SALL4 expression in melanocytes of healthy human skin, in agreement with a previous study[30], while SALL4 was strongly expressed in human melanoma tissue (Supplementary Fig. 2a, b).

**Conditional knockout of *Sall4* in the melanocytic lineage leads to reduced primary melanoma formation**. Given the re-expression of Sall4 in hyperplastic melanocytic lesions, we next addressed whether Sall4 is essential for melanoma formation. We therefore crossed the $Tyr::Nras^{Q61K}$; $Cdkn2a^{-/-}$ melanoma mouse model[31,32] with inducible $Tyr::Cre^{ERT2}$ [33] and $Sall4^{lox/lox}$ [34] and $R26R$-$LSL$-$GFP$ mice[35] (Fig. 2a). By doing so, we obtained a mouse model that spontaneously developed melanoma, in which Sall4 could be deleted in melanocytes upon tamoxifen (TM) injections (Fig. 2b) and Cre activity could be traced by GFP expression (Supplementary Fig. 3a). We assessed Sall4 expression levels in this model by mRNA analysis of isolated melanocytes from recombined postnatal day 8 mice of either wild-type ($Tyr::Cre^{ERT2}$ $LSL$-$R26R$-$GFP$ or $-tdTomato$), $Sall4^{+/-}$ conditional knockout (cko) ($Tyr::NRas^{Q61K}$ $Cdkn2a^{-/-}$ $Tyr::Cre^{ERT2}$ $Sall4^{lox/wt}$ $LSL$-$R26R$-$GFP$ or $-tdTomato$), or $Sall4^{-/-}$ cko ($Tyr::NRas^{Q61K}$ $Cdkn2a$ $^{-/-}$ $Tyr::Cre^{ERT2}$ $Sall4^{lox/lox}$ $LSL$-$R26R$-$GFP$ or $-tdTomato$) genotypes (Supplementary Fig. 3b). Hyperplastic melanocytes exhibited more than 10-fold increased Sall4 levels compared to wild-type melanocytes while Sall4 levels gradually decreased upon conditional knockout of Sall4 alleles, with $Sall4^{-/-}$ cko melanocytes having Sall4 expression levels comparable to wild-type melanocytes (Supplementary Fig. 3b). Interestingly, when we compared the numbers of primary melanomas in adult Sall4 control and $Sall4^{-/-}$ cko mice, we found that upon Sall4 ablation, primary tumor formation was impaired (Fig. 2c, d). Heterozygous $Sall4^{+/-}$ cko mice displayed no phenotype in comparison to control animals in regard to primary tumor numbers (Fig. 2c, d). However, $Sall4^{+/-}$ cko tumors were characterized by a decreased proliferation rate compared to Sall4 wild-type primary tumors (Fig. 2e; Supplementary Fig. 3c).

***Sall4* loss results in an increased metastatic burden in vivo**. In addition to proliferation, SALL4 has been associated with increased migration and invasion of solid tumors other than melanoma[36–38]. Therefore, we assessed whether loss of Sall4 affects melanoma metastasis formation in our melanoma-susceptible mouse model. For this purpose, we quantified the formation of lung micrometastases traced by GFP expression in $Sall4^{-/-}$ cko and $Sall4^{+/-}$ cko animals (Fig. 2f, g, h; Supplementary Fig. 4a, b). Melanoma identity of the micrometastases was confirmed by immunohistochemical staining for the melanocyte marker MITF (Fig. 2g). Strikingly, we found a significant increase in micrometastases counts in $Sall4^{-/-}$ cko as well as in $Sall4^{+/-}$ cko mice compared to the control animals (Fig. 2h). These data suggest that in melanoma—similar to other cancer types—Sall4 is essential for tumor growth, while its depletion or downregulation leads to increased micrometastases formation, which is in contrast to its function observed in other cancer types[36–38].

**SALL4 promotes proliferation and suppresses invasion in human melanoma cells**. To validate the findings from our mouse model in human melanoma cells, we carried out siRNA-mediated SALL4 knockdown experiments in various human melanoma cell lines with different mutational backgrounds. Namely, M010817

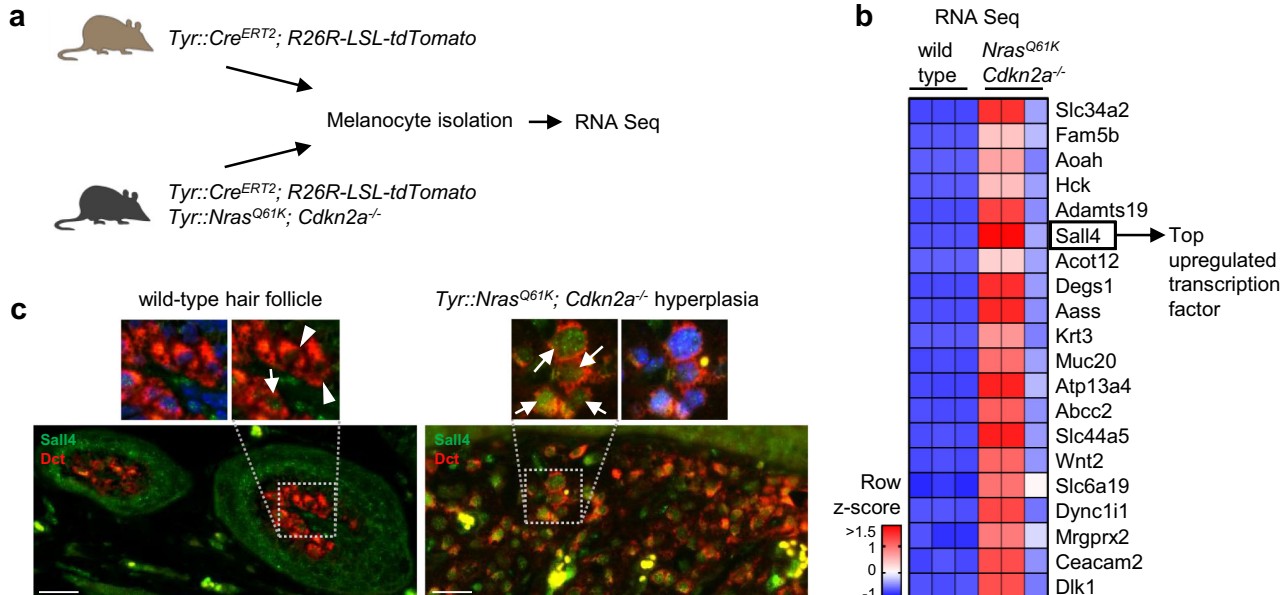

**Fig. 1 Sall4 is upregulated in hyperplastic melanocytes. a** Experimental scheme of isolated melanocytes for RNA sequencing either from *Tyr::CreERT2; R26R-LSL-tdTomato* wild-type mice or *Tyr::CreERT2; R26R-LSL-tdTomato; Tyr::NrasQ61K; Cdkn2a−/−* mice with hyperplastic, melanoma-prone skin. At the age of 4 months, the mice were injected on 5 consecutive days with 80 μg g−1 body weight tamoxifen to induce tdTomato expression in melanocytes. The hair growth cycle in wild-type animals was synchronized by dorsal hair plucking. Melanocytes were isolated from mouse back skin of wild-type and melanoma model mice (3–6 mice per sample) and FACS sorted for cKit (staining) and tdTomato (endogenous). **b** Row z-score heatmap of normalized counts from RNA sequencing as illustrated in **a**. Visualized are the top 20 most upregulated genes comparing hyperplastic (*Tyr::NrasQ61K; Cdkn2a−/−*) to wild-type melanocytes. Genes are ordered top to bottom with genes with highest LogFC values on top. Cutoffs were set at *p* values < 0.05 and FDR < 0.05. Sall4 (LogFC value = 9.59) represents the only and hence highest upregulated transcription factor within those top 20 upregulated genes. The complete list of all differentially expressed genes can be found under Supplementary Data 1. **c** Immunohistochemical staining of wild-type melanocytes in murine hair follicles (left panel) and melanoma-prone melanocytes in hyperplastic *Tyr::NrasQ61K; Cdkn2a−/−* mouse skin (right panel). Arrowheads point towards wild-type melanocytes with undetectable Sall4 expression; arrows point towards hyperplastic and rare wild-type melanocytes with detectable Sall4 protein. Experiment has been repeated independently three times with similar results. Scale bars 25 μm.

(*NRASQ61K*-mutated), M070302 (unknown mutation status), M150548 (*BRAFV600E*-mutated), MM150536 (*BRAFV600E*-mutated), WM1361A (*NRASQ61R* and *PTEN+/−*-mutated), and M121224 (*NRASQ61K*- and *BRAFV600E*-mutated) cells were subjected to different experimental setups to address aspects of proliferation, tumor growth, migration, and invasion (Fig. 3). After validation of knockdown efficiency of SALL4 on mRNA and protein levels (Fig. 3a–c), cell proliferation upon SALL4 knockdown was assessed using the xCELLigence® Real-Time Cell Analysis (RTCA) DP Instrument. The human melanoma cell lines tested displayed decreased proliferation capacities after SALL4 knockdown (Fig. 3d–f; Supplementary Fig. 5a, b). To test whether knocking down SALL4 would also lead to decreased tumor growth of human xenografts in vivo, we next injected human melanoma cells 24 h after siRNA transfection subcutaneously into nude mice and analyzed tumor growth within 6 days after grafting. In line with our previous findings, the majority of assessed siSALL4-treated human melanoma cells showed reduced growth capacities in vivo (Fig. 3g, h). Since our transgenic mouse model had indicated a role of Sall4 in tumor growth as well as in micrometastasis formation, we next analyzed melanoma cell migration and invasion in regards to SALL4 expression. First, to address whether cell intrinsic migratory capacities correlate with SALL4 levels, human melanoma cells were subjected to a Corning Transwell® migration assay (Supplementary Fig. 5c). Interestingly, we found that migratory cells from two of three cell lines tested expressed significantly less SALL4 than nonmigratory cells of the same cell lines (Supplementary Fig. 5d). Second, to assess whether reduced SALL4 expression functionally leads to increased invasiveness, human

melanoma cells treated with siSALL4 or siControl were seeded into Corning Transwell® invasion chambers wherein a porous membrane is coated with a matrigel layer through which the cells have to invade. Cells that had passed through the matrigel-coated membrane to the other side of the membrane (invasive cells) were stained with DAPI and analyzed. Consistent with our previous results, all three siSALL4-treated human melanoma cell lines tested showed a significantly increased invasion capacity compared to siControl-treated cells (Fig. 3i–k).

**SALL4 negatively regulates a set of melanoma-specific invasiveness genes.** To determine the molecular mechanism by which SALL4 loss leads to increased invasion, we performed RNA seq on siControl and siSALL4-treated M010817 cells and found 1004 genes significantly upregulated and 1140 genes significantly downregulated in siSALL4 over siControl samples (Fig. 4a; Supplementary Data 2). Gene ontology (GO) analysis with Meta-Core™ revealed that the upregulated genes were strongly enriched in Process Networks related to cell adhesion/cytoskeleton, developmental processes related to epithelial-to-mesenchymal transition, angiogenesis, and others (Fig. 4b; Supplementary Data 2). Oppositely, the downregulated genes were enriched in Process Networks related to cell cycle regulation, inflammation, and developmental processes such as hedgehog signaling or melanocyte differentiation (Fig. 4c; Supplementary Data 2). The differential gene expression obtained upon SALL4 knockdown pointed towards an upregulation of EMT-related genes and a downregulation of differentiation genes, overall suggesting acquisition of a transcriptional signature typical of melanoma cell phenotype switching. To assess whether there was a significant

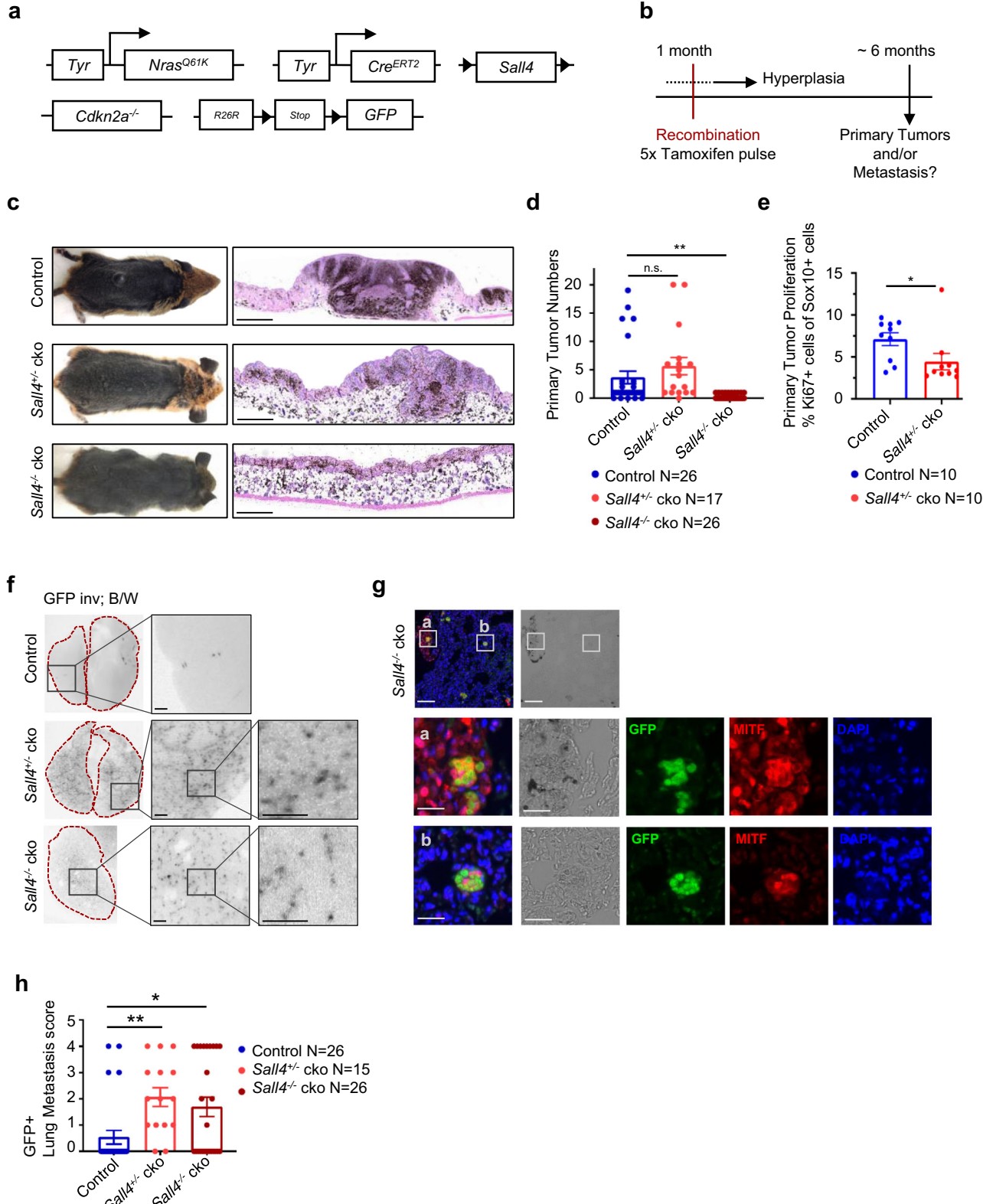

enrichment in genes related to phenotype switching among our differentially expressed genes upon siSALL4, we carried out Gene Set Enrichment Analysis (GSEA)[39], where the ranked siSALL4 signature was compared to published melanoma phenotype switching signatures. We found that SALL4-regulated genes showed a significant enrichment and positive correlation with published invasiveness signatures[40–42], while they significantly anti-correlated with their corresponding proliferation signatures[40–42] (Fig. 4d). Next, siSALL4-mediated upregulation of a set of invasiveness genes representing the top enriched Meta-Core™ Process Networks and also the published melanoma phenotype switching signatures (bold genes in Fig. 4b, c) was

**Fig. 2 Sall4 is essential for primary tumor formation, but its depletion leads to increased micrometastasis. a** Genetics scheme of the $Tyr::Nras^{Q61K}$; $Cdkn2a^{-/-}$ transgenic mouse model spontaneously developing melanoma, which was crossed with inducible $Tyr:Cre^{ERT2}$; $Sall4^{lox/lox}$; $R26R-LSL-GFP$ mice, allowing ablation of Sall4 from the melanocytic lineage upon tamoxifen administration. **b** Experimental scheme depicting how at 1 month of age the experimental mice from **a** undergo Cre-mediated recombination due to 5 consecutive i.p. tamoxifen injections. Hyperplasia gradually develops from birth of the pups. Primary tumors and metastasis were assessed at around 6 months of age. **c** Photographs of control and heterozygous ($Sall4^{lox/wt}$; termed $Sall4^{+/-}$ cko) and homozygous ($Sall4^{lox/lox}$; termed $Sall4^{-/-}$ cko) Sall4 cko animals (left panel). Hematoxylin and eosin staining of back skin from respective control, $Sall4^{+/-}$ cko or $Sall4^{-/-}$ cko animals (right panel), which has been repeated in independent experiments with similar results 7 times. Scale bars 500 μm. **d** Quantification of primary tumor numbers of control ($Sall4^{+/+}$ and non-tamoxifen-injected animals), $Sall4^{+/-}$ cko and $Sall4^{-/-}$ cko animals. **e** Quantification of proliferation rate, assessed by immunohistochemistry (see Supplementary Fig. 3c), in primary tumors of control and $Sall4^{+/-}$ cko animals. **f** Binocular images of mouse lungs. The endogenous fluorescent GFP signal was imaged under a fluorescent binocular and for visualization inverted and set to black/white (B/W). Dark spots therefore represent inverted GFP$^+$ spots set to B/W. Scale bars 500 μm. **g** Immunohistochemical stainings of mouse lung sections to verify melanoma identity of GFP$^+$ spots by means of MITF expression. Scale bars top panel 100 μm, second and third panel 25 μm. **h** Quantification of GFP$^+$ lung metastases of tamoxifen-injected control ($Sall4^{+/+}$), $Sall4^{+/-}$ cko, and $Sall4^{-/-}$ cko animals. Metastasis score = 0 indicates <5 GFP$^+$ lesions, 1 >5 lesions, 2 >20 lesions, 3 >50 lesions, 4 >100 lesions. In **d, e, h**, error bars represent mean ± SEM with N indicated in the respective figures. Two-sided $t$-tests between groups were performed for significance with $p$ values ≥0.05 = n.s.; <0.05 = *; <0.01 = **, with $Sall4^{-/-}$ cko in **d** $P = 0.0049$; $Sall4^{+/-}$ cko in **e** $P = 0.0456$; $Sall4^{+/-}$ cko in **h** $P = 0.0011$; $Sall4^{-/-}$ cko in **h** $P = 0.0129$. Source data for **d, e, h** are provided as a Source Data file.

validated by qRT-PCR in the five previously used human cell lines (Fig. 4e) and, for selected gene products, by immunocytochemistry and western blot in M010817 cells (Supplementary Fig. 6a, b). Together, our data suggest that the reduced expression of SALL4 induces human melanoma cell invasion via upregulation of known melanoma invasiveness genes, which could explain the increased metastasis burden seen upon Sall4 loss in the $Tyr::Nras^{Q61K}$; $Cdkn2a^{-/-}$ melanoma mouse model.

**SALL4 and HDAC2 interact and directly regulate a set of target genes.** SALL4 has been shown to regulate transcription by a variety of different mechanisms[26]. In mouse ESCs for example, Sall4 has been shown to exert stemness regulatory function via direct binding and activation of a distal enhancer of the gene $Pou5f1$, which encodes the pluripotency regulator Oct4[25]. Furthermore, in adult human CD34$^+$ hematopoietic stem cells, a whole set of direct targets of SALL4 has been identified by Chromatin Immunoprecipitation (ChIP)-chip and additional ChIP-qPCR validation[43]. Interestingly, neither $POU5F1$ nor the 16 validated genes (except for $HN1$) from Gao et al., (2013b) were significantly changed in our RNA sequencing upon SALL4 knockdown in melanoma cells. Similarly, $MYC$ has previously been shown to be a direct target of SALL4 in endometrial cancer[38], but we could not detect any altered $MYC$ expression upon SALL4 knockdown. We therefore hypothesized that in melanoma SALL4 might exert its regulatory function in an alternative manner and have a different array of target genes.

It has previously been reported that SALL4 can interact with epigenetic co-factors both in stem cells as well as in cancerous cells[26]. One type of epigenetic enzymes that has been shown to interact with SALL4 are histone deacetylases (HDACs), specifically HDAC1 and HDAC2[28,44,45], which are part of the Nucleosome Remodeling Complex (NuRD)[46]. Since HDAC1, HDAC2, and HDAC3 have been reported to be overexpressed in melanoma cells compared to primary melanocytes[47], we addressed whether in human melanoma cells, SALL4 can interact with one of the HDACs. By Co-immunoprecipitation (Co-IP) experiments we detected protein interaction between SALL4 and HDAC2 in the human melanoma cell line M010817 (Fig. 5a; Supplementary Fig. 7), which led to the hypothesis that SALL4 might repress invasiveness genes via recruitment of histone deacetylases, leading to epigenetic silencing of target genes such as invasiveness genes. To test this idea and address which genes are directly bound by both SALL4 and HDAC2, we carried out a cleavage under targets and release using nuclease (CUT&RUN) sequencing experiment for SALL4 as well as for HDAC2. Of note,

we chose to perform CUT&RUN with two different antibodies per factor, each set consisting of one antibody that had previously been published for CUT&RUN or ChIP sequencing plus one additional one (Supplementary Data 3). To determine target genes of both SALL4 and HDAC2, we performed peak calling with SEACR for loci that contained significant peaks with at least 3 of the 4 antibodies used (Fig. 5b, c, d; Supplementary Data 4). The '3of4' antibody approach allowed us on one hand to strengthen the specificity of the SALL4-HDAC2 targets and on the other hand to identify novel peaks that might only be recognized by one or another antibody due to epitope masking in protein complexes at specific locations. Interestingly, several of the identified SALL4-HDAC2 peaks were associated with putative invasiveness genes, with peaks either at the transcription start site (TSS) (such as for $VEGFR-1$) (Fig. 5e), downstream of the TSS (such as for $TGFBR2$) (Fig. 5f), or within annotated putative regulatory elements (such as for $PDGFC$, $CDH2$ (N-cadherin), and $FN1$) (Fig. 5g–i). The fact that these genes were upregulated upon SALL4 depletion (Fig. 4b, e) and direct targets of both SALL4 and HDAC2 is consistent with the idea that SALL4 recruits HDAC2 to these specific loci, resulting in histone deacetylation and hence repression of these invasiveness-related target genes.

Next, we analyzed with HOMER the DNA binding motifs of the CUT&RUN peaks that are unique for SALL4 (Supplementary Figs. 8, 9, 10), unique for HDAC2 (Supplementary Figs. 11, 12, 13), and most importantly, the DNA binding motifs that are shared between SALL4 and HDAC2 (Supplementary Fig. 14). Interestingly, we found amongst our top SALL4-HDAC2 shared de novo DNA binding motifs matches for known transcriptional regulators of the NC that are re-expressed and have functional implication in melanoma, such as SOX10 and SOX9[48] or RUNX1[49,50], and also key regulators of melanocyte differentiation and hence melanoma, such as MITF[51] or TFAP2C[52,53] (Fig. 5j). This further strengthens the hypothesis that SALL4 and HDAC2 together regulate melanocyte and melanoma-specific cellular processes.

**Differentially expressed SALL4-HDAC2 targets enrich in cell adhesion-related processes.** Since we had found that SALL4 and HDAC2 bind to a large set of common loci, we wanted to correlate their direct targets with differential expression upon SALL4 knockdown. As we hypothesized that SALL4 recruits HDAC2 to specific loci to repress gene activity, we assessed which SALL4-HDAC2 targets (peaks with at least 3 of 4 SALL4-HDAC2 antibodies) were significantly upregulated after SALL4 knockdown

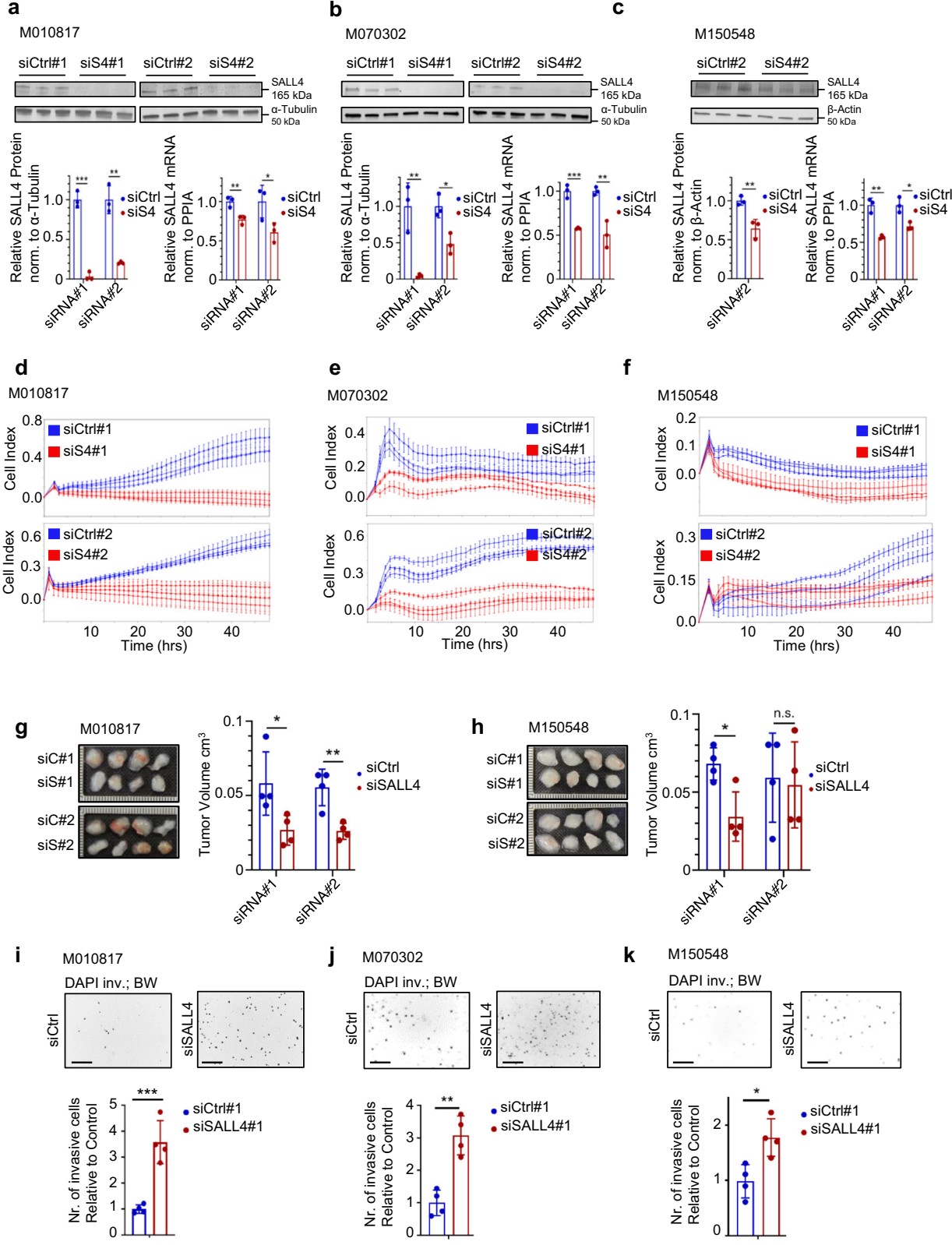

(RNA Seq) and found 184 direct SALL4-HDAC2 targets with increased expression upon SALL4 KD (Fig. 6a; Supplementary Data 4). MetaCore™ Process Network enrichment analysis revealed that these targets were associated with cell adhesion, TGFβ signaling, angiogenesis, and EMT (Fig. 6b; Supplementary

Data 4). For instance, among the SALL4-HDAC2 targets upregulated upon SALL4 knockdown, we found integrins (ITGB1, ITGA6, ITGA4), N-cadherin (CDH2), TGFBR2, VEGFR-1, PDGFC, LOXL2, MAPK8, and many others which are associated with invasive melanoma phenotypes[40–42] (Fig. 6b, Supplementary

**Fig. 3 SALL4 knockdown leads to reduced proliferation and increased invasion. a–c** SALL4 expression after 48 h knockdown with two different siRNAs assessed by western blot (top panel and bottom panel left (quantification)) and qRT-PCR (bottom panel right) on the same samples. Experiments performed in human melanoma cells with different mutational backgrounds: M010817 ($NRAS^{Q61K}$), M070302 (unknown mutational status), M150548 ($BRAF^{V600E}$). siS4: siSALL4. **d–f** xCELLigence real-time analysis of cell proliferation (Cell Index) of M010817, M070302, and M150548 cells after 48 h siRNA-mediated SALL4 knockdown as in **a–c**. Three samples per experimental group were analyzed, each consisting of four technical replicas (proliferation wells analyzed by xCELLigence). **g–h** In vivo xenografts after SALL4 knockdown: 24 h after siRNA treatment in vitro, cells were grafted subcutaneously onto nude mice and let grown for 6 days, when images were taken and tumor sizes of M010817 and M150548-derived grafts were quantified. Scale units = 1 mm. **i–k** Corning Transwell® invasion assay: siControl or siSALL4-treated cells (48 h) were seeded onto the porous, matrigel-covered membrane of an invasion chamber insert and let to migrate from FCS-free (top chamber) to FCS high (lower chamber) medium. Numbers of invaded cells on the bottom side of the invasion membrane were analyzed after 24 h. Invaded cells were stained with DAPI, imaged and set to black and white (top panels) for quantification (bottom panels). Scale bars 200 μm. Error bars represent mean ± SD and for significance, two-sided *t*-tests were performed with N = 3 (**a–f**) and N = 4 (**g–k**) and p values ≥0.05 = n.s.; <0.05 = *; <0.01 = ** and <0.001 = *** with siRNA#1 in **a** (lower left panel) P = 0.0001; siRNA#2 in **a** (lower left panel) P = 0.0015; siRNA#1 in **a** (lower right panel) P = 0.0099; siRNA#2 in **a** (lower right panel) P = 0.0476; siRNA#1 in **b** (lower left panel) P = 0.0070; siRNA#2 in **b** (lower left panel) P = 0.0133; siRNA#1 in **b** (lower right panel) P = 0.0006; siRNA#2 in **b** (lower right panel) P = 0.0058; siRNA#2 in **c** (lower left panel) P = 0.0094; siRNA#1 in **c** (lower right panel) P = 0.0016; siRNA#2 in **c** (lower right panel) P = 0.0129; siRNA#1 in **g** P = 0.0372; siRNA#2 in **g** P = 0.0047; siRNA#1 in **h** P = 0.0117; P in **i** = 0.0009; P in **j** = 0.0012; P in **k** = 0.0129. Source data for all panels are provided as a Source Data file.

Data 4). These results strongly support our hypothesis that SALL4 and HDAC2 co-repress invasiveness genes in melanoma and that either SALL4 or HDAC inhibition induces their expression.

We next analyzed the SALL4-HDAC2 target genes that were downregulated upon SALL4 loss (Supplementary Fig. 15a; Supplementary Data 4). Of note, MetaCore™ analysis showed that in general the Process Network enrichment of the siSALL4 downregulated direct SALL4-HDAC2 targets was less significant and processes were represented by fewer genes (Supplementary Fig. 15b) than was the case for the siSALL4 upregulated direct SALL4-HDAC2 targets (Fig. 6b). Nevertheless, of interest, the top enriched process of siSALL4 downregulated direct SALL4-HDAC2 targets was related to melanocyte differentiation, represented by downregulated direct targets such as MITF, DCT, β-catenin (CTNNB1), and tyrosinase (TYR), among others (Supplementary Fig. 15b). These data suggest that melanocyte differentiation genes, although they can be bound by SALL4-HDAC2, are subject to positive regulation by SALL4 by a mechanism that remains to be elucidated. Possibly, transcriptional activation in these cases might involve tertiary co-factors recruited to SALL4-HDAC2 target loci. To address this, we reanalyzed the SALL4-HDAC2 peaks within melanocyte differentiation genes (Supplementary Data 5) using CIIIDER to predict transcription factors (TFs) significantly enriched at these peaks (Supplementary Data 5). The resulting list of TFs (Supplementary Data 5) putatively bound to the same loci as SALL4 and HDAC2 was filtered for high stringency and further analyzed with STRING to predict protein–protein interactions with SALL4 (Supplementary Fig. 15c, red cluster). Interestingly, this in silico approach revealed TFs regulating NC development and melanoma as putative SALL4 binding partners at melanocyte differentiation gene loci with joint SALL4 and HDAC2 peaks such as TFAP2A (Supplementary Fig. 15c, red cluster), a well-known transcriptional activator in NC cells[54,55] and a regulator of melanocyte differentiation genes[53]. Hence, our data raise the possibility that amongst others, TFAP2A can be recruited to SALL4-HDAC2-target elements of melanocyte differentiation genes and that SALL4 (and possibly HDAC2) loss leads to a loss in TFAP2A-mediated transcriptional activation of these genes even if HDAC2-mediated repression is attenuated after SALL4 loss.

We also elaborated on genes that might putatively be regulated exclusively by either SALL4 or HDAC2 by screening for protein-coding genes that only show CUT&RUN peaks with the two SALL4 antibodies the two HDAC2 antibodies, respectively, but no 3 of 4 shared peaks (Supplementary Data 6). By doing so, we found that genes exclusively bound by SALL4 enrich in biological processes related to neurogenesis, neurophysiological processes, and others, while genes exclusively bound by HDAC2 enriched in various different process classes such as immune cell adhesion, cell cycle, apoptosis, and developmental processes (Supplementary Data 6). These data suggest that SALL4 largely relies on the interaction with HDAC2 to negatively regulate invasiveness genes in melanoma.

To functionally strengthen this hypothesis, we inhibited HDACs in human melanoma cell lines with HDAC inhibitors (HDACi) for 48 h and measured gene expression by qRT-PCR (Fig. 6c). Specifically, treatment with the class I HDACi inhibitor Mocetinostat, which inhibits HDACs 1, 2, and 3, resulted in a differential gene expression pattern similar to SALL4 knockdown, in that invasiveness genes were upregulated, while melanocyte differentiation genes were downregulated (Fig. 6c). A similar, although less striking effect was detected after treatment with the pan-HDAC inhibitor Panobinostat (Fig. 6c). Importantly, phenotypically HDACi treatment led to increased invasion in vitro for the majority of tested human melanoma cell lines (Fig. 6d, e).

These findings are in accordance with our results that showed increased invasiveness of melanoma cells upon SALL4 loss (Fig. 3i–k) and with previous studies that have reported HDAC inhibitor-induced invasiveness in melanoma cells and other cancer cells[56,57]. We therefore further addressed the expression of invasiveness genes in HDACi-treated melanoma cells in vivo. Human melanoma xenografts in athymic nude mice were treated with Mocetinostat and Panobinostat (Supplementary Fig. 16a) and tumor lysates were analyzed for expression of invasiveness genes at the experimental endpoint. While mice treated with either Panobinostat or Mocetinostat showed reduced xenograft tumor growth (Supplementary Fig. 16b), the expression of established invasiveness genes, such as AXL or NGFR was upregulated (Supplementary Fig. 16c). Moreover, we detected a trend of reduced expression of melanocyte differentiation genes, such as MITF, MLANA, or DCT, in the lysates of HDACi-treated tumors (Supplementary Fig. 16c).

**SALL4 knockdown leads to differential histone acetylation in invasiveness genes.** Given that SALL4-HDAC2 target genes belonging to invasiveness processes are upregulated upon SALL4 loss or HDAC inhibition, we wanted to assess whether SALL4 regulates epigenetic activation of invasiveness genes in general. Since HDACs catalyze deacetylation of histones and, consequently, transcriptional repression of target genes[58,59], we

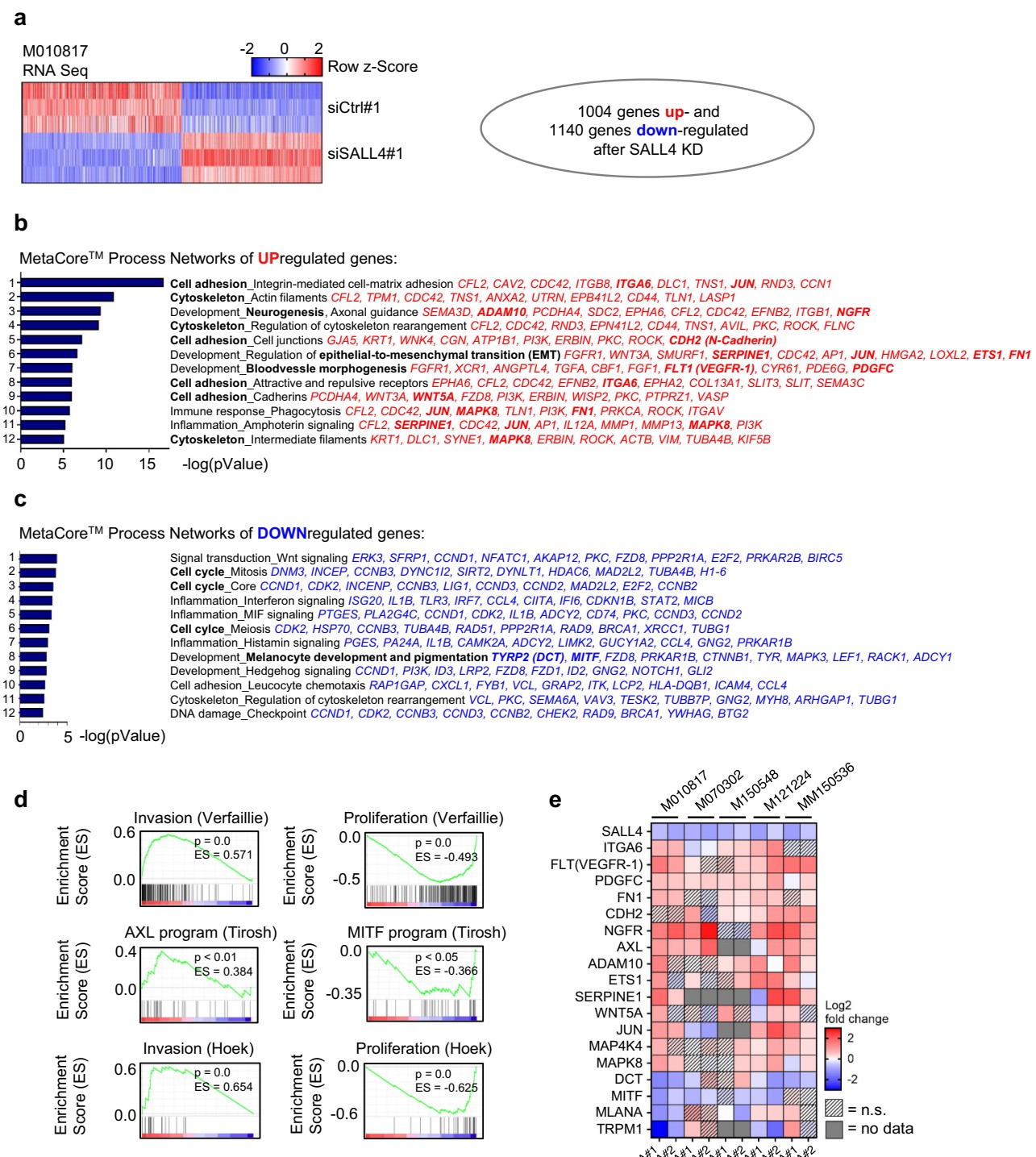

**Fig. 4 RNA sequencing of SALL4 knockdown reveals an invasiveness gene expression signature. a** RNA seq row z-score heatmap of differentially expressed (DE) genes after 48 h SALL4 knockdown in the human cell line M010817. Cutoffs were set at Log2 ratio ≥ 0.27 or ≤−0.27, p value < 0.05 and FDR < 0.05 and resulted in 1004 genes significantly upregulated and 1140 genes significantly downregulated. **b** Top 12 MetaCore[TM] process networks of upregulated genes. Top ten most upregulated genes per process are listed and genes validated in additional cell lines are highlighted in bold. **c** Top 12 MetaCore[TM] process networks of downregulated genes. Top ten most downregulated genes per process are listed and genes validated in additional cell lines are highlighted in bold. **d** Gene Set Enrichment Analysis (GSEA) of DE genes after SALL4 knockdown (log2 ratio-ranked) as in **a** with published genesets characterizing invasive versus proliferative melanoma cells[40-42]. **e** Log2 expression ratio heatmap of qRT-PCR-based gene expression analysis of specific genes of interest (normalized to PPIA and set relative to siCtrl-treated cells) in human melanoma cell lines (M010817: NRAS$^{Q61K}$; M070302: unknown mutational status; M150548: BRAF$^{V600E}$; M121224: NRAS$^{Q61K}$ and BRAF$^{V600E}$; MM150536: BRAF$^{V600E}$) after 48 h treatment with two different siRNAs targeting SALL4. Two-sided t-tests were performed with N = 3 and p values ≥ 0.05 = n.s. Source data for **e** are provided as a Source Data file.

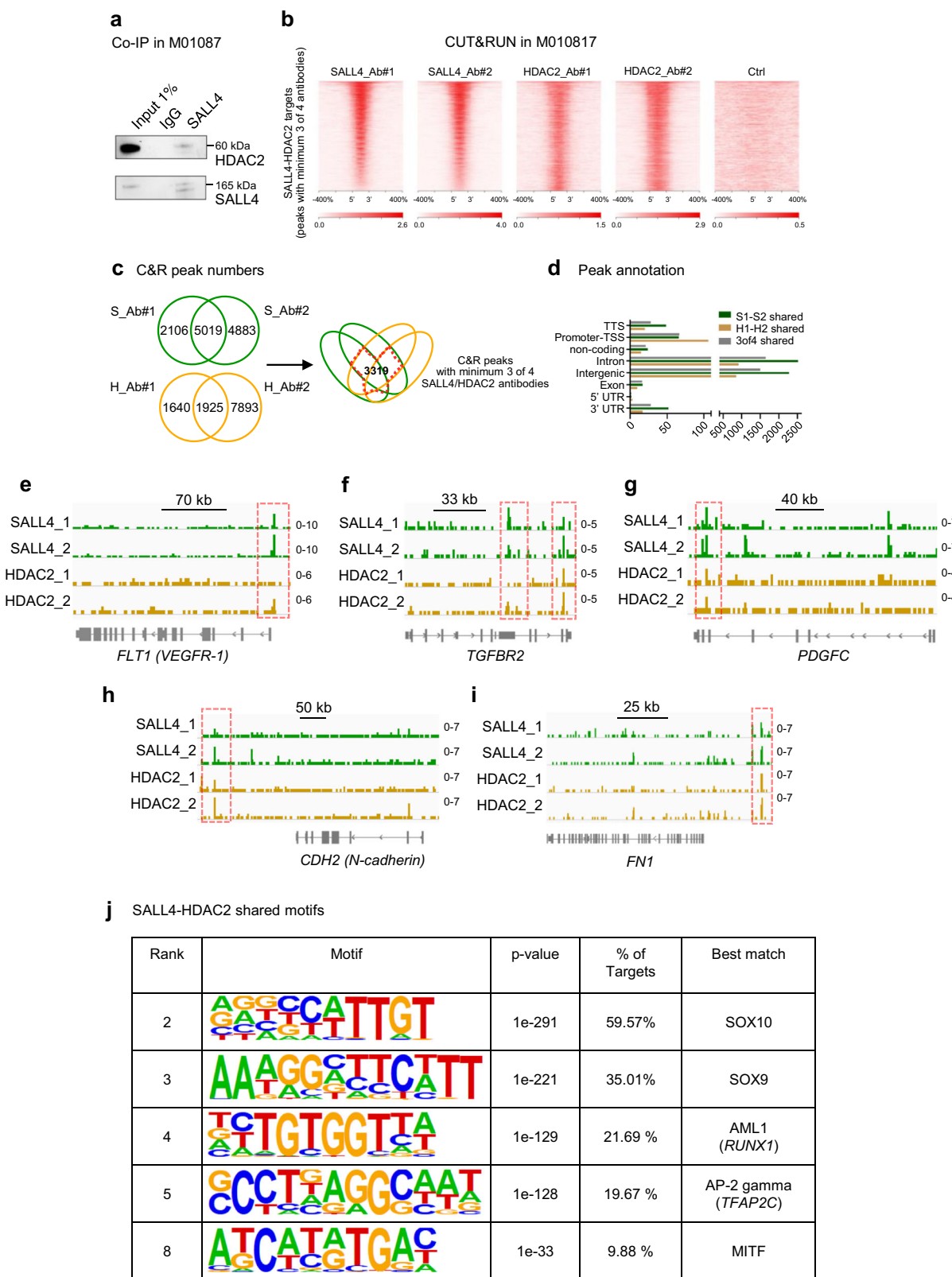

hypothesized that SALL4 knockdown would lead to derepression and increased acetylation of target genes associated with invasiveness. To address this hypothesis, we performed an H3K27ac Chromatin Immunoprecipitation sequencing (ChIP-seq) experiment of cells treated with siCtrl or siSALL4 (Supplementary Data 7) to identify genes with differential histone acetylation upon SALL4 knockdown (Fig. 7a, Supplementary Data 8). Overall, we found more gained H3K27ac peaks than lost ones upon SALL4 knockdown (Fig. 7a, b). Most of the peaks were annotated to intron and intergenic regions or promotor/

**Fig. 5 SALL4 and HDAC2 interact and have a set of common target genes in melanoma cells. a** Western blot for SALL4 and HDAC2 after Co-Immunoprecipitation (Co-IP) with a SALL4 antibody in the human melanoma cells M010817. Experiment has been repeated independently with similar results two times. **b** CUT&RUN (C&R) in M010817 cells with two antibodies against SALL4 (SALL4_Ab#1 (S_Ab#1) and SALL4_Ab#2 (S_Ab#2)) and two antibodies against HDAC2 (HDAC2_Ab#1 (H_Ab#1) and HDAC2_Ab#2 (H_Ab#2)) and Ctrl (anti-FLAG) visualized as read density heatmaps of the centered peaks (within 10 kb) for all loci showing peaks with at least 3 of the 4 (2x SALL4, 2x HDAC2) antibodies. **c** C&R peak numbers called with SEACR for single antibodies, shared between antibodies and shared between at least 3 of the 4 SALL4-HDAC2 antibodies. In total 3319 loci contained peaks for at least 3 of 4 antibodies tested and were found in total 2301 different genes. Those peaks were used for further analyses correlating the direct targets with either expression (Fig. 6a, b) or acetylation status (Fig. 7h). **d** Annotation of C&R peaks to genetic regions. Green: the 5019 SALL4_Ab#1 and SALL4_Ab#2 shared peaks; yellow: the 1925 HDAC2_Ab#1 and HDAC2_Ab#2 shared peaks; grey: the 3319 peaks shared between at least 3 of the 4 SALL4_Ab#1, SALL4_Ab#2, HDAC2_Ab#1, and HDAC2_Ab#2 antibodies. TTS: transcription termination site, TSS: transcription start site, UTR: untranslated region. **e–i** Specific gene tracks visualized with IGV. Green: SALL4_Ab#1 and SALL4_Ab#2; yellow; HDAC2_Ab#1, and HDAC2_Ab#2. Red dashed lines highlight significantly called peaks (with at least 3 of 4 antibodies). **j** Selected de novo DNA binding motifs of SALL4-HDAC2 (at least 3 of 4 antibodies) shared peaks analyzed by HOMER. Source data for **a** and **d** are provided as a Source Data file.

transcription start sites (TSS) (Fig. 7b), such as for the invasiveness genes *VEGFR-1*, *FN1*, *PDGFC*, and *NGFR* (promoter) or *AXL* (upstream), which showed significantly gained H3K27ac peaks (Fig. 7c–g, red bars and red, dashed highlighting) mediated by SALL4 knockdown, while no significantly lost H3K27ac peaks were found at the presented tracks.

Since we had hypothesized that increased histone acetylation in siSALL4 cells can partially be linked to reduced SALL4-mediated HDAC2 recruitment to specific target genes (i.e., invasiveness genes), we next wanted to address whether differential acetylation patterns can indeed be detected upon SALL4 knockdown at loci bound by both SALL4 and HDAC2. In line with our hypothesis, differential acetylation ChIP-seq peaks were present within 10 Kb intervals of the SALL4-HDAC2 peaks, as determined by means of a read density heatmap (Fig. 7h). Thus, H3K27ac is differentially regulated within a fraction of the SALL4-HDAC2-bound loci. Moreover, when we correlated the loci bound by SALL4-HDAC2 and showing increased H3K27ac ChIP-seq peaks in siSALL4 (Fig. 7h) with increased RNA expression after SALL4 knockdown (Fig. 4b, Supplementary Data 2), we again found, among others, invasiveness genes such as TGFBR2, ITGA6, or VEGFR-1 (Supplementary Data 9), that were enriched in biological processes related to cell adhesion, TGFβ signaling, and others (Supplementary Data 9).

Of importance, the heatmap correlating SALL4-HDAC2 targets with differential acetylation (Fig. 7h) also revealed that a large fraction of SALL4-HDAC2 peaks likely regulate more distant regulatory elements and not the exact same loci, to which the two proteins directly bind. This is to be expected due to the fact that any genomic locus that becomes functionally proximal to the SALL4/HDAC2 protein duet—which could also happen via genomic looping of far distant regulatory regions—could be differentially acetylated by HDAC2. Therefore, as it is difficult to define the exact genomic loci that the SALL4/HDAC2 complex regulates, we decided to investigate the functional impact of SALL4-recruited HDAC2 on a genome-wide level by correlating the regions of differential acetylation with the genes differentially expressed after SALL4 knockdown (Fig. 7i, Supplementary Fig. 17a). To do so, we restricted our H3K27ac ChIP-seq analysis to proximity to TSS by exclusively analyzing changed peaks within −15/+10 kb of the TSS. We found significantly gained H3K27ac peaks in 2566 genes and significantly lost H3K27ac peaks in 1131 genes (Supplementary Data 8). Next, we did an overlay of the genes with significantly gained or lost H3K27ac marks and increased or decreased transcription based on RNA seq, respectively. Within the more than 2000 genes with gained H3K27ac marks, 261 genes also showed increased mRNA levels (Fig. 7i, Supplementary Data 8), while within the more than 1000 genes with lost H3K27ac levels, 137 genes showed decreased mRNA expression (Supplementary Fig. 17a, Supplementary

Data 8). Interestingly, among the 261 genes with gained H3K27ac and increased transcription, we could again identify invasiveness genes such as VEGFR-1, PDGFC, ITGA6, NGFR, AXL, FN1, SERPINE1, among others, and MetaCore™ process network enrichment again resulted in pathways related to cell adhesion, EMT, angiogenesis, and others (Fig. 7j).

GSEA on the 398 genes with gained or lost histone acetylation and increased or decreased expression, respectively, (combined from Fig. 7i and Supplementary Fig. 17a) confirmed a positive correlation of the siSALL4 signature with the invasiveness signature of Verfaillie and colleagues (2015), while it showed a negative correlation with the corresponding proliferation signature (Fig. 7k), further suggesting that downmodulation of SALL4 changes the activating chromatin mark H3K27ac of genes related to phenotype switching.

Of note, while we hypothesize that SALL4 knockdown leads to derepression of invasiveness genes through attenuated HDAC2-mediated target gene repression, epigenetic activation of target genes after SALL4 loss could be boosted by direct targets of SALL4 and HDAC2 that function as epigenetic activators. For instance, we found the lysine demethylase 4 C (KDM4C) or the lysine acetyltransferase 2B (KAT2B) as direct targets of SALL4 and HDAC2 (Supplementary Fig. 18a, b), which have been identified as SALL4 targets in other systems as well[60,61]. Hence, upregulation of invasiveness genes after SALL4 depletion might be induced by a joint epigenetic mechanism of lost repression and gained activation of target genes.

To further test whether siSALL4-mediated upregulation of invasiveness genes was indeed dependent on differential acetylation, we combined SALL4 knockdown with an inhibitor of histone acetyl transferases (HATs), CTK7A, and quantified the expression of invasiveness genes. Of note, upregulation of invasiveness genes, such as ITGA6, FLT1 (VEGFR-1), PDGFC, FN1, NGFR, AXL, ETS1, and others, could be significantly rescued by addition of the HAT inhibitor CTK7A (Supplementary Fig. 19a-c), supporting our hypothesis that SALL4 regulates melanoma invasiveness genes via chromatin modulating mechanisms involving histone deacetylation.

## Discussion

*SALL4* is a member of the spalt-like (SALL) gene family, which are the vertebrate homologues of the developmental *spalt* (*sal*) genes in *Drosophila*[62]. Sall4 is a zinc finger transcription factor that plays an essential role in maintaining pluripotency and self-renewal of ESCs by regulating *Pou5f1* and by direct binding to Nanog and Oct4[25,63]. Sall4 is vastly expressed during murine embryonic development and of note, in a previous study we have detected its expression also in murine neural crest stem cells[19]. In the adult, however, Sall4 is mostly absent and detectable only in the germ cells of the ovaries or testis[64]. Similarly to mice, SALL4

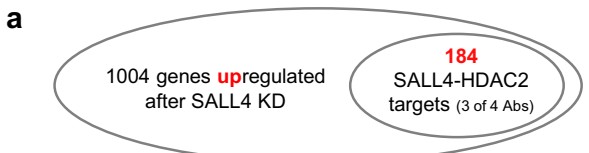

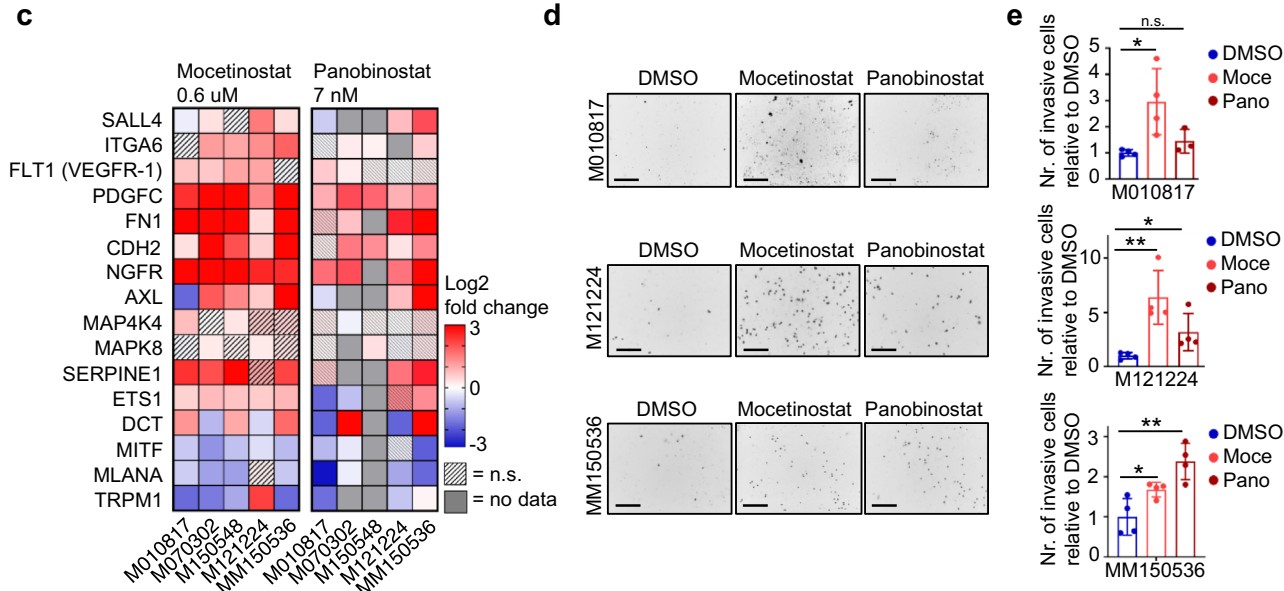

**Fig. 6 SALL4-HDAC2 targets that are upregulated after SALL4 depletion or HDAC inhibition enrich in invasion-related processes. a** Significantly upregulated genes after SALL4 knockdown (Fig. 4a, b) were overlaid with the C&R data of direct SALL4-HDAC2 targets (Fig. 5b, c), which resulted in 184 direct SALL4-HDAC2 targets significantly upregulated after SALL4 knockdown. **b** MetaCore™ Process Network enrichment of the 184 direct targets from **a**. The top 12 most significant processes are given with the specific genes of each process listed in red. Genes that were validated for DE in other cell lines are highlighted in bold. **c** Log2 fold change heatmap of qRT-PCR-based gene expression (normalized to PPIA and set relative to vehicle (DMSO)-treated samples) after 48 h cell treatment with the HDAC inhibitors Mocetinostat (0.6 μM) and Panobinostat (7 nM). t-tests were performed for significance between HDACi and vehicle-treated groups with N = 3 and p values ≥ 0.05 = not significant (n.s.). **d** Images of DAPI-stained nuclei in invasion assays as in Fig. 3 i–k. Cells were seeded into the invasion chamber, which was supplemented with vehicle or 600 nM Mocetinostat or 7 nM Panobinostat in both the starvation as well as the FCS high medium. After 24 h, the invaded cells were analyzed by counting DAPI-stained nuclei of invaded cells at the bottom of the membrane (inverted and set to black and white (B/W)). Scale bars 200 μm. **e** Quantification of invaded cells in **d** set relative to control cells. Moce: mocetinostat, Pano: panobinostat. Error bars represent mean ± SD in **e**. For significance, two-sided t-tests were performed with N = 3 (**c**) and N = 4 (**e**) and p values ≥0.05 = n.s.; <0.05 = *; <0.01 = **; and <0.001 = *** with Moce in **e** (top panel) P = 0.0218; Moce in **e** (middle panel) P = 0.0051; Pano in **e** (middle panel) P = 0.0463; Moce in **e** (lower panel) P = 0.0332; Pano in **e** (lower panel) P = 0.0051. Source data for **c** and **e** are provided as a Source Data file.

expression in adult human tissue is thought to be restricted to testis and ovaries[64] and hematopoietic stem/progenitor cells (HSPCs)[43].

Several studies have reported the re-expression of the stem cell factor SALL4 in the adult in different cancer types[26]. Mostly, the re-expression of SALL4 was associated with increased tumor cell proliferation and decreased patient survival. Here, we show that Sall4 is upregulated in hyperplastic murine melanoma-prone melanocytes and that its expression is essential for melanoma

primary tumor growth. However, in contrast to previous studies on other cancer types, our results reveal that depletion of *Sall4* in a murine melanoma model leads to increased micrometastasis formation while preventing sustained tumor growth. In line with these in vivo results, knockdown of SALL4 in human melanoma cells leads to reduced tumor cell proliferation and to the upregulation of a set of well-known melanoma invasiveness genes, inducing an invasive cell phenotype. Hence, our findings identify SALL4 as a negative regulator of melanoma cell

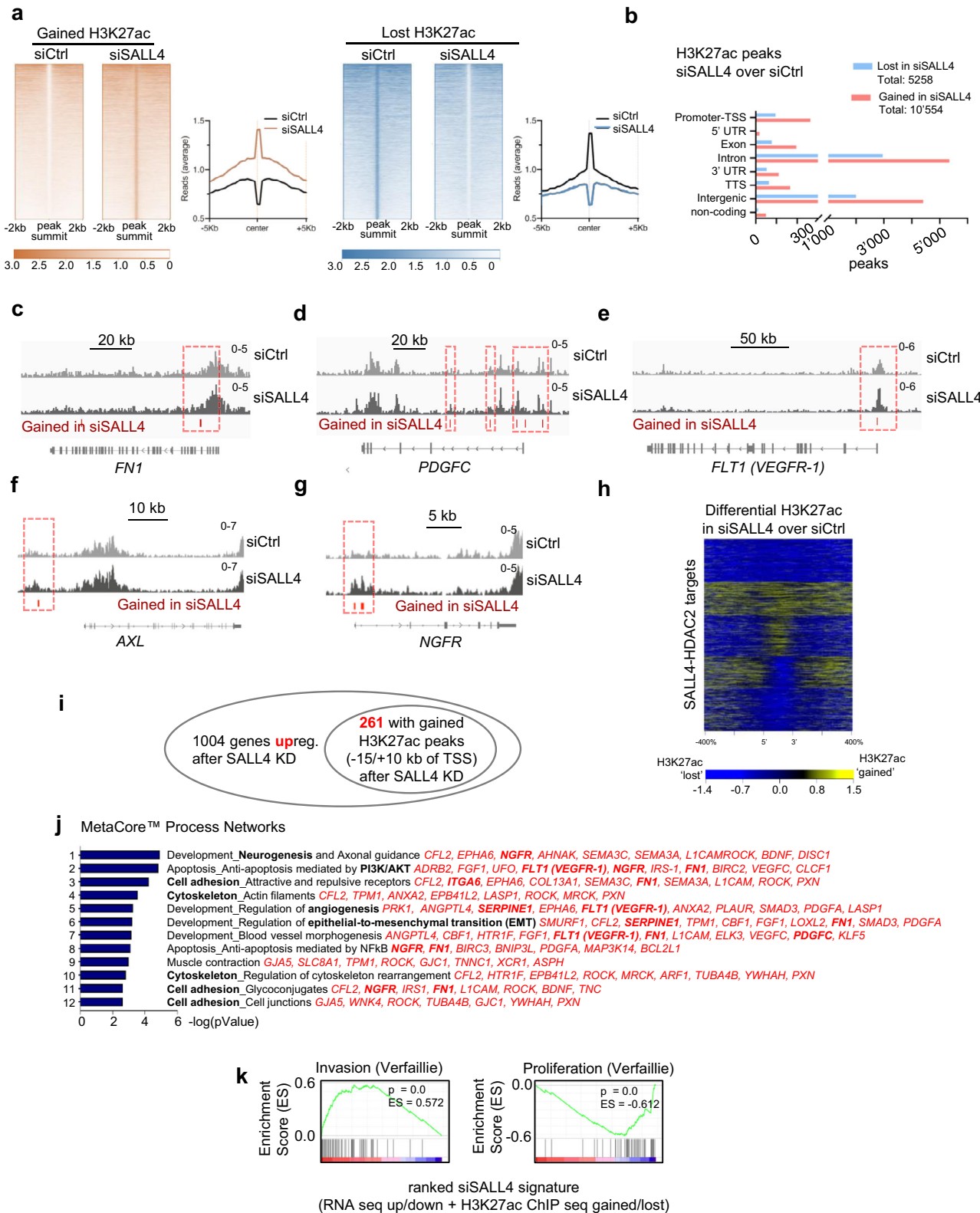

phenotype switching, i.e. the reversible change from a high proliferative/low invasive to a low proliferative/high invasive cell state[23,24,40]. Notably, SALL4 negatively controls melanoma invasiveness-related genes, as for instance NGFR, a potent regulator of phenotype switching[22], FN1, VEGFR-1, CDH2 (N-cadherin), and other genes implicated in melanoma invasiveness.

The binding of SALL4 to chromatin and epigenetic modifier enzymes and the resulting epigenetic rewiring of target genes has previously been shown in ESCs as well as in cancer. Amongst the published epigenetic co-factors of SALL4 are DNA methyltransferases (DNMT-1, DNMT-3A, DNMT-3B, and DNMT-3L), histone demethylase LSD1/KDM1A, and others[26]. Additionally,

**Fig. 7 SALL4 can regulate invasiveness genes through an epigenetic mechanism. a** Read density heatmap of gained and lost H3K27ac ChIP sequencing peaks (±2 kb) upon SALL4 knockdown (left panels) and average read distribution of gained and lost H3K27ac peaks within ±5 kb from peak center (right panels). **b** Distribution of the gained and lost H3K27ac peaks in different genetic regions. TTS: transcription termination site, TSS: transcription start site, UTR: untranslated region. **c–g** Representative tracks of invasiveness genes with significantly gained (red bars; highlighted by red, dashed lines) H3K27ac peaks in siSALL4 over siCtrl (no significantly lost H3K27ac peaks were detected for the visualized gene tracks). **h** Read density heatmap of differential H3K27ac ChIP-seq peaks (within 10 kb) in siSALL4 over siCtrl at direct target genes of SALL4-HDAC2 (CUT&RUN peaks with at least 3 of 4 SALL4/HDAC2 antibodies). **i** Significantly upregulated genes after SALL4 knockdown (Fig. 4a, b) were overlaid with those genes that have significantly gained H3K27ac marks −15/+10 kb of TSS (**a**, ocher panel). This resulted in 261 genes with activating chromatin marks that were at the same time upregulated after SALL4 knockdown. **j** MetaCore™ Process Network enrichment of the 261 genes from **i**. The top 12 most significant processes are indicated with the specific genes of each process listed in red. **k** Gene Set Enrichment Analysis (GSEA) of the combined 261 upregulated (**i**) and 137 downregulated genes (Supplementary Fig. 17a) with differential acetylation after SALL4 knockdown (ranked according to log2 expression ratio of RNA seq results) with the previously published melanoma programs of Verfaillie and colleagues (2015). Source data for **b** are provided as a Source Data file.

based on tandem mass spectrometry studies on ESCs and 293 T cells overexpressing SALL4, it has been shown that SALL4 can also bind to the NuRD complex members HDAC1 and HDAC2[45]. Here we show that SALL4 can build a protein complex together with HDAC2 also in human melanoma cells and that SALL4 and HDAC2 together directly bind to genes involved in melanocyte and melanoma biology, such as VEGFR-1, CDH2 (N-cadherin), FN1, TGFBR2, MITF, and others.

Strengthening the hypothesis that SALL4-HDAC2 epigenetically repress transcriptional activity of invasiveness genes, knockdown of SALL4 leads to increased H3K27 acetylation and increased transcription of melanoma invasiveness-related genes, such as FN1, VEGFR-1, PDGFC, and NGFR, which could be partially rescued by administration of a histone acetyltransferase (HAT) inhibitor. Likewise, HDAC inhibition with two different HDAC inhibitors, leads to upregulation of a similar set of invasiveness-related genes and to increased invasiveness in vitro. This supports the hypothesis that SALL4 inhibits invasiveness-related genes in melanoma via interaction with HDACs. HDAC inhibitors have been considered for usage in combinatorial therapies for several cancers including melanoma[47,65–67]. Our data suggest caution in the use of these inhibitors because of potential adverse, metastasis-promoting effects of HDAC inhibitors in melanoma patients.

Others have shown different sets of target genes epigenetically silenced by SALL4. Lu et al (2009), for instance, identified by ChIP the SALL4 targets SALL1 and PTEN. Since the binding sites of those target genes were co-occupied by NuRD components with HDAC activity, the authors argued that SALL4 silences PTEN and SALL1 by interacting with NuRD. Phenotypically, the decreased expression of the tumor suppressor PTEN in SALL4 transgenic mice was associated with myeloid leukemia and cystic kidneys[45]. Indeed, in our experiments, binding of SALL4 and HDAC2 to PTEN (but not SALL1) was confirmed by CUT&RUN (Supplementary Data 3 and 4). However, we did neither observe differential acetylation patterns in ChIP-seq nor transcriptional changes in RNA seq of PTEN or SALL1 upon SALL4 knockdown. In addition, other previously identified direct targets of SALL4[43] have not come up in our analyses either, which suggests that SALL4 might regulate a set of specific targets in melanoma cells.

Regarding melanoma disease progression, there is increasing evidence that the re-expression of NCSC-related factors can regulate melanoma initiation and later stages of the disease, such as resistance to therapies or invasion and metastatic spread[16]. However, unlike melanoma cells undergoing phenotype switching, NCSCs during embryonic development continue to proliferate when they migrate and invade distant tissues[68]. Intriguingly, while some NCSC factors, such as SOX10 and YY1, are activated upon melanoma formation and are required for melanoma growth[16,19,29,69], other NCSC-associated factors, such as CD271/NGFR/p75[NTR], PAX3, and FOXD3 promote melanoma cell invasiveness and metastasis formation[22,70]. This suggests that the embryonic program active in NCSCs segregates in melanoma to regulate distinct aspects of phenotype switching, namely proliferation vs. invasion. Our study adds SALL4 to the growing list of stem cell factors known to control melanoma cell biology[16]. Similar to SOX10 and YY1, SALL4 is upregulated in melanoma cells and is essential for primary tumor formation and proliferation. Importantly, however, SALL4 depletion in melanoma leads to increased invasiveness and micrometastasis formation. This is reminiscent of melanoma cells displaying reduced SOX10 levels, which is associated with increased expression of SOX9, another factor involved in NC development, and increased invasiveness[48,71]. Thus, it will be important to elucidate to what extent the gene regulatory program of NCSCs is functionally implicated in mediating cellular plasticity during melanoma disease progression. Knowledge of this program might allow defining strategies targeting both tumor growth and metastasis formation.

## Methods

**Transgenic mice and in vivo TM application.** *Tyr::Nras*[Q61K] animals and *Cdkn2a*-deficient mice have been described previously[31,32]. Also the *Tyr::Cre*[ERT2] murine line[33], *Sall4*[lox] mice[34], and *R26R-LSL-GFP* mice[35] have been analyzed previously. Mice were bred and crossed in-house to generate the *Tyr::Nras*[Q61K]; *Tyr::Cre*[ERT2]; *Sall4*[lox/lox]; *R26R-LSL-GFP* genotype and resulted in a mixed genetic background. All animal breeding, housing and experimentation was conducted according to the guidelines of the veterinary office of the Canton of Zurich, Switzerland. Specifically, animals were housed in a controlled environment with a 12 h light/dark cycle, with free access to water and food and at temperatures of 21–23 °C and humidity of 40–60%. Genotyping was performed on toe or ear biopsies, followed by PCR on isolated DNA using the Taq PCR Core Kit (201225, Qiagen) and primers as listed in Supplementary Table 2. For conditional ablation of *Sall4*, 8-weeks-old transgenic mice of both genders were injected intraperitoneally (i.p.) with 100 μl tamoxifen (TM) (T5648, Sigma–Aldrich) diluted in ethanol and sunflower oil (1:9 ratio) at a concentration of 1 mg d⁻¹ for 5 days according to an established protocol[29]. Melanoma-developing mice were monitored and euthanized at an endpoint defined by adverse clinical symptoms, such as multiple skin tumors ∅ >5 mm, weight loss (△m> 15%) or a hunched back. All animal experiments have been approved by the veterinary authorities of Canton of Zurich, Switzerland and were performed in accordance with Swiss law.

**Quantification of skin melanomas and metastases.** At sacrifice of the animals, skin melanomas and metastasis were assessed. Above a diameter of 2 mm (∅ > 2 mm), developing trunk skin lesions were considered as melanomas. The tumors of the heterozygous cko group were verified to have been recombined and lost *Sall4* by either GFP or Sall4 immunohistochemical stainings of tumor sections. Animals with non-recombined tumors were excluded from the analysis. The Control group consisted of noninjected *Tyr::Nras*[Q61K] *Cdkn2a*[−/−] *Tyr::Cre*[ERT2] *Sall4*[lox/lox] *R26R::GFP* and TM-injected *Tyr::Nras*[Q61K] *Cdkn2a*[−/−] *Tyr::Cre*[ERT2] *Sall4*[wt/wt] *R26R::GFP* animals for analysis of primary tumor numbers and of TM-injected *Tyr::Nras*[Q61K] *Cdkn2a*[−/−] *Tyr::Cre*[ERT2] *Sall4*[wt/wt] *R26R::GFP* animals for analysis of lung metastases. To assess whether GFP⁺ spots in the mouse lungs were of melanoma identity, MITF and GFP co-stainings were performed on histological sections of the mouse lungs. For quantification of micrometastasis, the number of GFP⁺ spots in the lungs were categorized as follows: <5 lesions = score 0; >5 lesions = score 1; >20 lesions = score 2; >50 lesions = score 3; >100 lesions = score 4. Group sizes are indicated for each experiment in the respective figures.

**Melanocyte isolation from neonate murine skin and qRT-PCR.** For isolation of reporter-traced melanocytes, recombination of neonate mice was achieved by injecting the breast-feeding mother i.p. with TM as described before[29] one day after birth of the offspring and for 5 consecutive days. Tissue of neonate mice was harvested as follows. Mice were euthanized by decapitation and trunk back skin was dissected, cut into small pieces and incubated in DMEM/F12 (Gibco 21041025) with 10% FCS and 1.5 mg/ml Collagenase Type I (Merck SCR103) and 1:1000 Protector RNase Inhibitor (Roche 3335399001) for 1 h at 37 °C on a shaker. The digested tissue was next spun down at 1200 rpm for 5 min, and the pellet resuspended in complete DMEM/F12. To make a single-cell suspension, the solution was next pushed three times through an 18 G needle (Sterican 4665120) and next, five times through a 21 G Needle (Sterican 304432). The single-cell suspension was then filtered through a 40 μm cell strainer and centrifuged with $400 \times g$ for 5 min at 4 °C. The pellet was resuspended in DMEM/F12 with 10% and 1 mM EDTA and 1:1000 RNase Inhibitor and cells sorted by means of FACS on a BD FACSAria™ III Cell Sorter (BD Biosciences). Sorted cells were next subjected to RNA isolation performed with the Direct-zol™ RNA MiniPrep Kit (ZYMO Research, R2050) according to the manufacturer's instructions. A total of 500 ng isolated RNA was reverse transcribed using Maxima First Strand cDNA Synthesis Kit (K1641, Thermo Scientific) according to the manufacturer's recommendations. Real-time quantitative PCR (qRT-PCR) was performed on a LightCycler 480 System (Roche) using LightCycler 480 SYBR Green I Master (4707516001, Roche) with technical triplicates for each sample.

**Immunohistochemical stainings.** After dissection, mouse tissue samples were washed in PBS (Thermo Fisher, 10010) and fixed in 4% Roti Histofix (Carl Roth, P087.3) overnight at 4 °C. Next, samples were embedded in paraffin and cut into sections of 5 μm. Samples were then deparaffinized and either stained for hematoxylin and eosin according to standard procedures or stained for immunofluorescence as follows. Sections were subjected to microwave-based antigen retrieval using in-house made Citrate Buffer. After washing slides with PBS, they were blocked for one hour at room temperature in blocking buffer containing 1% BSA (Sigma–Aldrich, 05470) and 0.2% Tween (Sigma–Aldrich, P1379) in PBS. After washing in PBS, primary antibodies were applied in blocking buffer at the concentrations indicated in Supplementary Table 3 and incubated at 4 °C overnight. After washing the sections again in PBS, they were incubated with secondary antibodies in blocking buffer for 1 h at room temperature (RT). Finally, Nuclei were stained with Hoechst 33342 (Sigma–Aldrich, 14533) and slides were mounted with Fluorescent Mounting Medium (Dako, S3023). Immunohistochemical and fluorescent sections were imaged with either the Mirax Midi Slide Scanner (Zeiss) or a DMI 6000B microscope (Leica). Murine hyperplastic skin and primary tumors in Supplementary Fig. 1 and human healthy melanocytes and primary melanoma in Supplementary Fig. 2 were stained accordingly: After deparaffinization of the sections as described above, an antigen retrieval step was performed using citrate buffer (S2369, Dako). Prior to blocking, the endogenous peroxidase activity was quenched using 3% Hydrogen Peroxide Solution for 1 h at RT and the sections were permeabilized and blocked in blocking buffer (1% BSA in PBS and 0.05% Triton X-100) for 15 min at RT. The Sall4 staining was amplified using tyramide superBoost Kit (Thermo Fisher, Cat# B40926). As secondary antibodies, a biotin-coupled antibody (Jackson Immuno Research, 711-065-152, 1:100) and streptavidin-HRP (Jackson Immuno Research, 016-030-084, 1:100) were used. Multiplexing was performed by stripping off the Sall4 primary antibody. The sections were immersed in citrate buffer (S2369, Dako) and stripping was performed using the HistosPRO machine (Milestone, SW2.0.0) for antigen retrieval at 100 °C for 20 min. The sections were blocked another time in blocking buffer for 15 min at RT prior to application of the Sox10 homemade antibody in blocking buffer overnight at 4 °C. Samples were mounted in Fluorescent Mounting Medium (Dako, S3023) and imaged using Zeiss Axio Scan. Z1.

**Human cell lines.** Cells used for experiments were either provided by the University Hospital Zurich where surplus tumor material was obtained after surgical removal of melanoma metastases from patients after written informed consent and approved by the local IRB (EK647 and EK800) or commercially acquired (A375 and WM1361A). Medical staff at the University Hospital Zurich confirmed the melanoma diagnosis of the tumor material by histology and immunohistochemistry. The selective adherence method of Raaijmakers et al. (2015)[72] was used to establish primary melanoma cell cultures from patient biopsies. All primary melanoma cell lines are also included in the URPP biobank, University Hospital Zürich, Department of Dermatology. Cells were grown and cultured for experiments in RPMI 1640 (Sigma Life Science, USA) supplemented with 10% fetal bovine serum (Gibco, Life Technologies, USA), 4 mM l-Glutamine (25030, Life Technologies), and a mix of penicillin–streptomycin antibiotics (15070, Life Technologies). Work with human melanoma cells was approved by the local ethical review board (KEK Nr. 2014-0425). Detailed information about the cell lines can be found in Supplementary Table 1.

**siRNA transfection.** Cells were cultured in starvation medium (0.5% FCS) and after 24 h transfected at 60% confluency in complete growth medium (RPMI 1640 with 10% FCS, 4 mM L-Glutamine, Pen-Strep) with a final siRNA concentration as indicated in Supplementary Table 4. For transfection, the JetPrime transfection kit (114, Polyplus transfection) was used according to the manufacturer's guidelines. Further information on siSALL4#1 and #2 can be found in Supplementary Table 4.

**RNA isolation and real-time PCR.** RNA extraction and DNase treatment of samples was performed using the ReliaPrep™ RNA Miniprep kit (Promega, Z6010) according to manufacturer's guidelines. Purified RNA was quantified using nanodrop and subjected to reverse transcriptase reaction using Maxima First Strand cDNA Synthesis Kit (K1641, Thermo Scientific) according to manufacturer's recommendations. Real-time quantitative PCR (qRT-PCR) was performed on a LightCycler 480 System (Roche) using LightCycler 480 SYBR Green I Master (4707516001, Roche). Primers used for qRT-PCR are listed in Supplementary Table 2. Each sample was analyzed in technical triplicates on the LightCycler 480 device (Roche) and relative quantified RNA was normalized to housekeeping gene transcripts as indicated for each figure.

**Protein isolation and western blotting.** Cultured cells were lysed and protein extracted as described previously[22]. Protein concentrations were determined with the BCA Protein Assay Kit (#23227, Thermo Fisher Scientific) using a DTX 880 Multimode Detector at 562 nm. Thirty micrograms of total protein were run through Mini-PROTEAN® TGX™ Precast Gels (Bio-Rad) and transferred onto nitrocellulose membranes (Bio-Rad, 2895), which were stained with primary antibodies in Odyssey blocking buffer (LI-COR Biosciences, 927–40000) overnight at 4 °C and visualized using secondary antibodies in Odyssey blocking buffer for 45 min at RT . Blots were scanned and quantified with an Odyssey imaging system (LI-COR Biosciences).

**xCELLigence real-time cell proliferation analysis.** For assessment of cell proliferation, we made use of the xCELLigence® Real-Time Cell Analysis (RTCA) DP Instrument (ACEA Biosciences). This system allows impedance-based real-time growth/proliferation measurements of adherent cells by measuring the net adhesion/confluency (Cell Index) of cells to high-density gold electrodes on custom-designed plates. The assay was performed according to the manufacturer's guidelines. Specifically, 48 h after transfection of the cells with the respective siRNAs (as described above), 20'000 cells were seeded into xCELLigence proliferation plates (E-Plate 16 PET; ACEA Biosciences, 300600890) in complete growth medium and monitored over 48 h with a measurement every 15 min. Four wells were measured for each sample as technical replicas. The average Cell Index for each sample plus standard deviation was calculated and plotted (for one measurement per hour over 48 h) by using the RTCA Software 2.1.0 (ACEA Biosciences).

**Human xenografts in immunocompromised mice.** Nude mice (Hsd:AthymicNude-Foxn1nu) were purchased from Harlan and housed under standard conditions with free access to water and food at temperatures of 21–23 °C and humidity of 40–60 %. Experiments were carried out with female mice of 6–10 weeks of age. Xenografts of human melanoma cells were generated by dissociating cultured cells with PBS + 2 mM EDTA to generate single-cell suspensions, which were resuspended in 100 μl of RPMI-1640 medium and mixed 1:1 with Matrigel matrix (356234, BD Biosciences). A total volume of 200 μl of the cell-matrigel mix was injected subcutaneously per injection site with a 1 ml syringe with a 25-gauge hypodermic needle. For xenografts of siRNA-treated M010817 and M150548 cells, 1'000'000 tumor cells were subcutaneously injected per injection site and let grown for 6 days. For the generation of xenografts that were treated with HDAC inhibitors, 300'000 M010817 cells were grafted subcutaneously per injection site (two injections per mouse). Fourteen days after cell injections, mice carrying human xenografts were treated with 10 mg kg$^{-1}$ body weight Panobinostat or 40 mg kg$^{-1}$ body weight Mocetinostat or vehicle every second day for 2 weeks. In vivo HDACi treatment was performed similarly to previously established protocols[73–75]. Specifically, Mocetinostat (MGCD0103; Selleckchem, S1122) and Panobinostat (LBH589; Selleckchem, S1030) were diluted at 100 mg ml$^{-1}$ and 25 mg ml$^{-1}$, respectively in DMSO (Sigma–Aldrich, D2650). Next, 10 μl of the drugs dissolved in DMSO were diluted with 90 μl sunflower oil and the total volume of 100 μl was injected i.p. for each treatment. Vehicle injections consisted of 10 ul DMSO (Sigma–Aldrich, D2650) diluted in 90 μl sunflower oil. Tumor length and width were measured with a caliper and xenograft volumes calculated according to the formula $V = \frac{2}{3}\pi x(\frac{a+b}{4})^3$, as previously published[76], wherein $a$ (mm) was the length and $b$ (mm) was the width of the tumor. Tumor xenografts (max ≤ 1 cm$^3$ or at experimental endpoint) were harvested from euthanized mice and for qRT-PCR-based gene expression analysis further processed as previously described for RNA isolation from murine lungs[22]. Briefly, dissected tumors were submerged in 500 μl of TRIzol® reagent (15596026, ThermoFisher) and homogenized with a tissue homogenizer (Polytron PT 2100, Kinematica) and RNA extracted following the manufacturer's guidelines. After extraction, 50 μg of RNA were purified and treated with DNAse, respectively, with the RNeasy Mini Kit (74104, Qiagen) and the RNase-Free DNase Set (79254, Qiagen). The reverse transcription and qRT-PCR was further done, as described above.

**Corning® Transwell® migration assay.** The migration assay was carried out similarly to the manufacturer's guidelines. Specifically, prior to seeding the cells into the transwell migration chamber they were incubated in starvation medium (RPMI + 1% FBS, P/S, L-Glut) for 24 h. Next, 50,000 cells in 500 μl FBS-free starvation medium were seeded per well onto the porous membrane of Clear Transwell Inserts (Corning, 3464) and placed within a multi-well plate with 800 ul of normal growth medium (RPMI + 10% FBS + P/S + L-Glut) in the bottom of the well. Cells in the transwell chambers were then incubated at 37 °C for 16 h. At the experimental endpoint, inserts were taken out and remaining starvation medium in the upper chamber was carefully pipetted up and down and then collected with the nonmigratory cells in suspension, centrifuged, and cell pellets further processed for RNA extraction. The medium with nonmigratory cells of three wells was pooled to make one sample. Next, membranes now free of nonmigratory cells were cut out of the flipped transwells and three membranes were collected in one Eppendorf to make one sample and were further processed with lysis buffer to extract RNA of migratory cells. RNA extraction and qRT-PCR were performed as described above.

**Corning® Matrigel® invasion assay.** The invasion assay was carried out similarly to the manufacturer's guidelines. Specifically, prior to seeding of the cells into the invasion chamber, they were incubated in starvation medium (RPMI + 1% FBS, P/S, L-Glut) for 48 h. On the day of cell seeding, invasion plates with inserts (Corning, 354480) were equilibrated by adding 800 ul blunt medium into the wells and 500 ul into each insert and incubated for 2 h at 37 °C. Cells were then collected, washed once in PBS, resuspended in starvation medium and seeded into each insert in a final volume of 500 ul. Of the cell line M010817 200,000 cells were seeded, of M070302 160,000 and of M150548 200,000 cells, respectively. Inserts were then placed into wells where blunt medium was aspirated and replaced by 800 ul of growth medium (RPMI + 10% FBS + P/S + L-Glut) and seeded cells were incubated for 24 h at 37 °C. For analysis, the medium in the inserts was aspirated and the membrane cleaned carefully but vigorously with cotton swabs and washed once with PBS. Cells were then fixed with 4% Roti Histofix (Carl Roth, P087.3) for 15 min at room temperature and washed once with PBS. Inserts were then carefully cut out with a scalpel and placed downside up onto microscopy glass slides, covered with some drops of Fluorescent Mounting Medium (Dako, S3023) with 1:1000 Hoechst 33342 (Sigma–Aldrich, 14533) and covered with glass cover slips. Image acquisition of the membranes with the invaded cells was done on a DMI 6000B microscope (Leica). Cell counts were determined using ImageJ. Tile scans of whole membranes were first set to 8 bit and inverted. The background was subtracted using default settings of 50 and the threshold was adjusted (auto) and under binary converted to a mask. Waterstedding was applied and a circle drawn (and saved as ROI for subsequent analyses), wherein particles were analyzed with 50–700 pixel size and 0.7–1.0% circularity. Same ROIs were used for analysis of membranes within one experiment. Four technical replicas i.e. membranes were used per experimental group. For HDAC inhibitor treatment in the invasion assay, the drugs were added to the starvation medium in the top chamber as well as the FCS high growth medium in the lower chamber for the whole duration of the assay at 600 nM (Mocetinostat) and 7 nM (Panobinostat) final concentrations. For siRNA treatment in the invasion assay, cells were transfected with the respective siRNA in vitro in normal growth medium (10 %) FCS. 24 h later, medium was exchanged to starvation medium (1%). Again 24 h later, siRNA-treated cells were seeded into the invasion chamber as described above and analyzed again 24 h later.

**RNA sequencing.** RNA sequencing of wild-type and hyperplastic murine melanocytes was performed in the frame of the study of Varum et al. (2019)[19] at the iGE3 Genomics Platform Geneva, Switzerland. RNA quantity and integrity were assessed with a Bioanalyzer (Agilent Technologies). Total RNA of 2 ng as input was used for reverse transcription and cDNA amplification using the SMART-Seq v4 kit from Clontech according to the manufacturer's instructions. 200 pg of cDNA were then used for library preparation using the Nextera XT kit from Illumina. Library molarity and quality was assessed with the Qubit and Tapestation using a DNA High sensitivity chip (Agilent Technologies). Libraries were pooled at 2 nM and loaded for clustering on 1 lane of a single-read Illumina flow cell. Reads of 50 bases were generated using the TruSeq SBS chemistry on an Illumina HiSeq 4000 sequencer. The reads were mapped with STAR aligner v.2.5.3a to the UCSC Mus musculus mm10 reference genome and differential expression analysis was performed with the statistical analysis R/Bioconductor package edgeR v.3.14.0. Supplementary Data 1 contains the complete list of differentially expressed genes in hyperplastic melanocytes compared to wild-type ones (with cutoffs for Log2 ratio ≥ 0.58 or ≤ −0.58, p value < 0.05 and FDR < 0.05), of which the top 20 genes are presented in Fig. 1b.

RNA sequencing of human melanoma cells with SALL4 knockdown was performed at the Functional Genomics Center Zurich, Switzerland. Specifically, total RNA of three experimental replicates of siControl and siSALL4 was isolated using the RNAeasy Kit (74104, Qiagen) and RNase-Free DNase Set (79254, Qiagen) according to the manufacturer's protocol. On the Agilent RNA ScreenTape assay and the Agilent 4200 TapeStation, quality control of total RNA was performed. With magnetic beads poly-A mRNA was enriched (TruSeq RNA Library Prep Kit v2) before cDNA synthesis and library preparation. After sequencing on the Illumina HiSeq 4000 platform, RNA counts were quantified from single-end reads using STAR aligner and mapped to the homo sapiens hg38

reference genome. Subsequent bioinformatic analysis of differentially expressed genes was performed with DeSeq2 version 1.16.1. Gene ontology network analysis was performed with MetaCore™ (Thomson Reuters). Supplementary Data 2 contains the complete list of differentially expressed genes (with cutoffs for Log2 ratio ≥ 0.27 or ≤ −0.27, p value < 0.05 and FDR < 0.05) used for the row z-score heatmap and the MetaCore™ analysis (also included in Supplementary Data 2) in Fig. 4 a, b.

**Immunocytochemistry.** For stainings of adherent cells, culture medium was aspirated from culture wells, cells washed with PBS (Thermo Fisher, 10010) and fixed in 4 % Roti Histofix (Carl Roth, P087.3) for 15 min at room temperature. Histofix was removed and cells washed three times for 5 min with PBS. Sections were blocked for 50 min at room temperature in blocking buffer containing 1% BSA (Sigma–Aldrich, 05470) and 0.2% Triton™ X-100 (Sigma–Aldrich, 9002-93-1) in PBS. After aspirating the blocking buffer, primary antibodies were applied in blocking buffer at the concentrations indicated in Supplementary Table 3 and incubated for 1.5 h at room temperature. After washing the sections again 2× with PBS, they were incubated with secondary antibodies in blocking buffer for 45 min at room temperature. Cells were again washed 2× with PBS and finally, nuclei were stained with Hoechst 33342 (Sigma–Aldrich, 14533) and slides were mounted with Fluorescent Mounting Medium (Dako, S3023). Sections were imaged with a DMI 6000B microscope (Leica).

**Gene Set Enrichment Analysis (GSEA).** GSEA[39] version 4.1.0 was run if there was a minimum of 15 shared genes between the datasets to compare and on default settings with 1000 permutations.

**Co-immunoprecipitation.** Co-immunoprecipitation experiments were performed using the Dynabeads Co-Immunoprecipitation Kit (Thermo Fisher, 14321D) and following the manufacturer's instructions. In brief, 1.5 mg Dynabeads M-270 Epoxy per sample were initially washed with 1 ml of C1 solution. The beads were collected using a magnet and then resuspended in 55 μl C1 solution + 20 g antibody (either SALL4, abcam #ab29112, or IgG, abcam # ab6709 or Cell Signaling # 2729 S) and equal volume of C2 solution. The beads were then coupled overnight at 37 °C. The next day, the beads were washed with HB and LB solution (always +0.01% Tween-20) respectively, and finally washed with and resuspended in SB solution (as in manufacturer instructions).10⁶ M010817 cells were harvested, weighted and resuspended in Extraction buffer in a ratio 1:9 (+cOmplete, Mini, EDTA-free Protease Inhibitor, Sigma #11836170001). Cells were lysed for 15 min on ice, centrifuged at $2600 \times g$ for 5 min at 4 °C and the supernatant was transferred to a new tube. The antibody coupled beads were then washed with 900 μl of Extraction buffer, collected with a magnet and gently resuspended in the cell lysate. Coupled beads and cell lysate were incubated for 30 min at 4 °C on a rotator. The beads were then washed for six times with the Extraction buffer, incubated 5 min at RT in LWB buffer (+0.01% Tween-20) and finally 5 min at RT in elution buffer (EB). The supernatant was then used to perform western blots and either 1% input or 0.01% input were used as positive control for the pulldowns of SALL4 (1%) or of HDACs (0.01%).

**Cleavage under targets and release using nuclease (CUT&RUN) sequencing.** M010817 cells were harvested by adding EDTA to the cell medium to a final concentration of 2 mM and by detaching cells by gentle scraping and pipetting. Cells were then washed three times by pelleting at $300 \times g$ for 5 min and resuspending in 2 ml wash buffer (20 mM HEPES pH 7.4, 150 mM NaCl, 2 mM Spermidine, EDTA-free protease inhibitor) each time. Cells were counted, and 500,000 cells per sample were bound to 40 μl of Concanavalin A beads prepared according to a previously established protocol[77]. Beads were divided into 2 ml tubes placed on a magnetic rack. Beads were allowed to bind the magnet until clear (~2 min), whereafter the supernatant was removed, and beads resuspended in 150 μl of antibody buffer (20 mM HEPES pH 7.4, 150 mM NaCl, 2 mM Spermidine, EDTA-free protease inhibitor, 0.05% Digitonin, 2 mM EDTA). Antibodies (HDAC2: Abcam ab12169 and Cell Signaling 57156 S; SALL4: Abcam ab29112 and Antibodies-Online ABIN6132627; and FLAG: Sigma–Aldrich F1365) were added at a 1:100 dilution to the respective samples. Cells were incubated overnight on a rocking table at 4 °C. Next, cells were washed two times by binding beads to magnetic rack, allowing to clear and resuspending in 1 ml digitonin-wash buffer (20 mM HEPES pH 7.4, 150 mM NaCl, 2 mM Spermidine, EDTA-free protease inhibitor, 0.025% Digitonin, 2 mM EDTA). Beads were then allowed to bind until clear (~2 min), whereafter the supernatant was removed and beads resuspended in 100 μl of 0.5× CUTANA pA/G-MNase (EpiCypher) solution (20 mM HEPES pH 7.4, 150 mM NaCl, 2 mM Spermidine, EDTA-free protease inhibitor, 0.025% Digitonin, 2 mM EDTA, 0.5× CUTANA). Next, the cells were incubated for 2,5 h on a rocking table at 4 °C and then washed two times by binding the beads to a magnetic rack, allowing to clear and resuspending in 1 ml digitonin-wash buffer. Beads were allowed to bind until clear (~2 min), whereafter the supernatant was removed and beads resuspended in 100 μl of digitonin-wash buffer. Next, tubes were submerged in ice-water and allowed to chill for ~1 min before 2 μl of 100 mM CaCl₂ was added to each sample. To stop the reaction, 100 μl STOP buffer (150 mM NaCl, 20 mM EDTA, 4 mM EGTA, 0.025% Digitonin, 100 μg/ml RNAse

A, 50 μg/mL Glycogen, 100 pg/ml yeast spike-in DNA) was added to each tube. Then, the tubes were placed in a 37 °C heat block for 30 min to release soluble chromatin from cells. Beads were again bound to the magnetic rack and allowed to clear completely, and the supernatant was transferred carefully into a DNA-lo bind tube.

DNA was extracted using Phenol:Chloroform extraction and libraries were prepared using KAPA HyperPrep Kit (Roche) using KAPA DUI adapters (Roche). Libraries were pooled, and later sequenced on a NextSeq 500 using a NextSeq 500/550 High Output Kit v2.5 (75 Cycles), generating 36 bp paired-end FASTQ files. Reads were trimmed using BBDuk, removing overrepresented repeat sequences (i.e., [TA]$_{18}$, [G]$_{36}$), artifact, and adapter sequences. Reads were aligned to the human genome (hg38) using bowtie[78] with settings -X 700 -m1 -v 3. Duplicate reads were removed, and files were sorted using samtools[79]. Mapped reads were filtered for size, keeping only reads with a fragment size at or below 120 base pairs. Bedgraph files were generated using bedtools genomecov, and peaks were called using SEACR version 1.3 (https://seacr.fredhutch.org/)[80], in relaxed mode, normalizing to the negative control. Single-gene tracks were visualized with the Integrative Genome Viewer (IGV) version 2.8.13 (https://software.broadinstitute.org/software/igv/download).

**HDAC inhibitor treatment in vitro**. With Mocetinostat (MGCD0103; Selleckchem, S1122) and Panobinostat (LBH589; Selleckchem, S1030) stocks of 10 mM drug in DMSO (Sigma–Aldrich, D2650) were generated and stored at −20 °C. For working concentrations of 600 nM Mocetinostat and 7 nM Panobinostat, respectively, stock solutions were further diluted in complete growth medium and added to cells in culture for 48 h.

**Chromatin immunoprecipitation of H3K27ac and sequencing**. ChIP analysis was performed as previously described[81]. Briefly, 1% formaldehyde was added to cultured cells to crosslink proteins to DNA. Isolated nuclei were then lysed and sonicated using a Bioruptor ultrasonic cell disruptor (Diagenode) to shear genomic DNA to an average fragment size of 200 bp. Twenty micrograms of chromatin was diluted to a total volume of 500 μl with ChIP buffer (16.7 mM Tris–HCl, pH 8.1, 167 mM NaCl, 1.2 mM EDTA, 0.01% SDS, 1.1% Triton X-100) and precleared with 10 μl packed Sepharose beads for 2 h at 4 °C. Precleared chromatin was incubated overnight with the indicated antibodies. The next day, Dynabeads protein-A were added and incubated for 4 h at 4 °C. After washing, bound chromatin was eluted with the elution buffer (1% SDS, 100 mM NaHCO$_3$). Upon proteinase K digestion (50 °C for 3 h) and reversion of crosslinking (65 °C, overnight), DNA was purified with phenol/chloroform, ethanol precipitated and quantified by Qubit$^{TM}$ dsDNA HS assay kit (Invitrogen). Library prep and sequencing was done at the Functional Genomics Center Zürich (FGCZ) with 1–10 ng of total DNA. Libraries were generated at the FGCZ and sequencing was performed on Illumina HiSeq2500. All sequence reads were mapped to the homo sapiens hg19 UCSC reference genome using Bowtie2 (http://bowtie-bio.sourceforge.net/bowtie2/index.shtml). Peak calling was performed with the HOMER tool package version 4.11 (http://homer.ucsd.edu/homer/) to find peaks in siCtrl and siSALL4 samples and significantly gained or lost H3K27ac peaks in the siSALL4 versus siCtrl sample. Supplementary Data 7 contains the list of significant peaks in both the siCtrl and siSALL4 samples and Supplementary Data 8 contains the significantly gained and lost H3K27ac regions of siSALL4 over siCtrl. Read density heatmaps for Figure 7a were generated with HOMER and single tracks visualized with the Integrative Genome Viewer (IGV) web application (https://igv.org/app/).

**HAT inhibitor (CTK7A) treatment in vitro**. With the Histone Acetyl Transferase Inhibitor VII CTK7A (Calbiochem, 382115) stocks of 10 mg ml$^{-1}$ drug in DMSO (Sigma–Aldrich, D2650) were generated and stored at −20 °C. For working concentrations of 50 μM CTK7A, stock solutions were further diluted in complete growth medium and added to cells in culture for 48 h.

**HDAC activator (ITSA-1) treatment in vitro**. With the HDAC activator ITSA-1 (Santa Cruz, CAS 200626-61-5) stocks of 25 mg ml$^{-1}$ drug in DMSO (Sigma–Aldrich, D2650) were generated and stored at 4 °C. For working concentrations of 100 μM ITSA-1, stock solutions were further diluted in complete growth medium and added to cells in culture for 48 h.

**Statistical analysis**. All statistical evaluations (unpaired, two-tailed Student's t-tests) were done using GraphPad Prism 5.0 and Excel with $p$ values > 0.05 = n.s., $p$ values ≤ 0.05 = *, $p$ values ≤ 0.01 = **, $p$ values ≤ 0.001 = ***. Experiments were done in number of replicates as indicated for each figure.

**Reporting summary**. Further information on research design is available in the Nature Research Reporting Summary linked to this article.

## Data availability

The RNA seq datasets of hyperplastic and wild-type melanocytes from Varum et al. 2019 are available on the European Nucleotide Archive (ENA) under accession code PRJEB30285. The RNA seq data of siControl and siSALL4-treated M010817 cells

generated for this study are available on ENA under the accession code PRJEB39208. The ChIP-seq data of H3K27ac in siControl and siSALL4-treated M010817 cells generated for this study are available on ENA under accession code PRJEB39209. The CUT&RUN seq data of SALL4 and HDAC2 in M010817 cells generated for this study are available on ArrayExpress under accession code E-MTAB-10163. The lists of differentially expressed genes from RNA seq experiments or lists of called peaks from ChIP-seq and C&R seq that resulted from our analysis of the deposited raw data are provided as Supplementary Data with this paper. Source data are provided with this paper.

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

## Acknowledgements

We thank the Functional Genomics Center Zürich (FGCZ) for support with RNA sequencing and ChIP sequencing for this manuscript in the frame of projects p2155 and p2419, respectively. We further thank Khanh Huynh for assistance with mouse experiments and Simon Söderholm for support with the ChIP-seq data analysis. This work has been supported by the University of Zurich (University Priority Research Program (URPP) Translational Cancer Research to M.L., R.D., K.B., S.V., and L.S. and Candoc Forschungskredit FK-19-026 to J.D.), by the Swiss National Science Foundation (31003A_169859 and 310030_192075 to L.S.) and the Swiss Cancer Research foundation (KFS-4570-08-2018 to L.S.). C.C. is a WCMM fellow supported by the Knut and Alice Wallenberg Foundation and by Cancerfonden (CAN 2018/542).

## Author contributions

J.D., A.B., and L.S. designed the study. J.D., A.B., M.P., D.D., J.H., and S.S. performed experiments and analyzed data. J.D., A.B., M.P., P.C., L.L., S.V., L.St., K.B., R.S., and C.C. analyzed data and provided intellectual guidance in experimental designs and interpretation of results. M.T. provided the *Sall4^{lox/lox}* mouse. R.D. and M.L. provided melanoma patient-derived cell cultures. J.D. and L.S. wrote the manuscript.

## Competing interests

The authors declare no competing interests.
