## [Peer Review File · Nature Communications]

REVIEWER COMMENTS

Reviewer #1 (Remarks to the Author):

Diener et al. study the role of the Stem cell factor SALL4 on melanomagenesis and melanoma metastatic spread. The authors combine genetic mouse model, transcriptomic analysis and in vitro assays in human melanoma cells to conclude that SALL4 expression regulates melanoma initiation and cell proliferation while its loss induces an undifferentiated, pro-metastatic melanoma phenotype similar to the previously published invasive AXLhigh/MITFlow phenotype. Furthermore the authors postulate that this phenotype change is mediated by epigenetic mechanisms that involve HDAC mediated histone modifications.

While there are some points to be raised, in general the data is clear and solid and the title reflects the conclusions the Diener et al. take from their results.

1. Figure 1b. Validation of SULL4 mRNA by qRT-PCR expression should be presented.
2. Figure 1c. If possible images with better resolution should be presented.
3. Figure 2 shows that heterozygous KO of SALL4 doesn't affect the number of tumors. What's the proliferation/mitotic rate in these tumors compared to SALL4 "wt" animals?
4. Figure 2d. What is the expression of SALL4 in +/+, +/- and -/- tumors at the mRNA level?
5. Figure 3a-f: Partial knock-down of human SALL4 by siRNA clearly affects proliferation, would this contradict the results obtained in Figure 2, regarding heterozygous SALL4 KO? This point should be addressed (see point 3).
6. Since the mouse model assesses melanoma formation in a mutNRAs background, I think the authors could have assessed in several mutRAS cell lines (WM1361, WM1366, SBCL2...). At present the authors only used a "pure" NRAs cell line (M01) to assess proliferation, migration and invasion. Still it would be interesting to assess if SALL4 regulates proliferation and invasion independently of the driver mutation (BRAF v NRAS) regulating melanoma cell biology.
7. Does SALL4 regulate an undifferentiated phenotype in mutBRAF human melanoma cells? If that is the case it would open an interesting hypothesis regarding the role of SALL4 in the regulation of mutBRAF melanoma cell response to MAPK inhibitors. Is SALL4 differentially expressed in melanoma tumors treated with BRAF/MEK inhibitors? I would advise the authors to perform such studies to strengthen the relevance of the manuscript.
8. Page 7, lines 135-141. Apparently tumors were analyzed "within 6 days after grafting". Yet these tumors reached a volume of 0.05cm³/5000mm³. Is there an error in the text or in the graph scale in Figure 3h?
9. Figure 4C. Validation of mRNA levels from relevant genes should be included.
10. Figure 5a. A blot showing the levels of immunoprecipitated SALL4 needs to be presented in the main figures. The same applied to Suppl. Figure S3a.
11. Figure 5f. What are the levels of MITF in these tumors?

Reviewer #2 (Remarks to the Author):

The manuscript by Diener, Baggiolini et al., explores the potential role of the stem cell factor SALL4 on the phenomenon of phenotype switching in melanoma. The authors found that Sall4 is elevated in hyperplastic murine melanocytes using a well-characterized GEMM and that its loss reduces primary tumor formation while increasing metastatic burden. The authors then study SALL4 function in human melanoma cells, and propose that this transcription factor (TF) regulates melanoma phenotype switching by suppressing an invasive gene signature through histone deacetylase recruitment.

The manuscript presents robust phenotypic data and the generation of a Sall4 KO mouse melanoma model is an important achievement that might be beneficial for further studies. However, the epigenetic mechanism by which SALL4 regulates gene expression has only been partially addressed and the SALL4-HDAC2 mechanistic link is not supported by solid experimental data. Other issues include cherry-picking of gene lists without providing a full view of the data and the lack of Sall4 ChIP-seq.

Therefore, while an interesting study, several important shortcomings must be addressed as described below.

Major points:

1. Sall4 depletion in the mouse models promotes metastasis and the mouse data is compelling. The authors then examine Sall4 function in human melanoma cells. In Figure 3 the authors perform siRNA-mediated knockdown in human melanoma cells and validate their findings from the mouse model with multiple phenotypic assays. However, the efficiency of SALL4 knockdown by mRNA expression is poor in all three melanoma lines, in particular in M12 cells where it barely reaches 30% reduction (Figure 3a). In contrast, Western blot analysis of SALL4 knockdown in M01 cells appears to totally deplete SALL4 (Supplementary Figure S2A). This discrepancy needs to be clarified. SALL4 protein levels upon knockdown should be assessed in all three melanoma cell lines and presented in the main figure to better correlate SALL4 knockdown levels with the different degrees of phenotypic changes observed. In addition, Figure S2A shows how SALL4 levels are much lower in melanoma cells than in the control embryonal carcinoma cell line NTERA2D1. What is the expression level of this protein in normal human melanocytes? This data would help validating the theory of SALL4 specific re-expression in melanoma tumors.

The authors reach Sall4 through RNA-seq data presented in Figure 1, but they don't provide the full picture and only present top 20 genes. They go on to state that Sall4 is the top TF, but don't provide any evidence of other TFs. Moreover, they only examine hyperplastic state of melanocytes. How do Sall4 levels compare in the melanomas that eventually develop in these mice? If its expression is elevated in hyperplastic state, does this correlate with proliferation of these cells?

2. The change in chromatin state caused by SALL4 depletion is key to the mechanistic value of this study. However, the mechanism by which SALL4 recruits HDAC2 on specific gene targets is not clear. In fact, the author's conclusions about SALL4's mechanism of action are based on the indirect evidence that histone acetylation is impaired in SALL4 knockdown cells. This is a substantial concern that needs to be address with different lines of experimental data as suggested here:

i) The Co-Immunoprecipitation experiment demonstrating the interaction between SALL4 and HDAC2 is a crucial point in the manuscript (Figure 5a). However, this assay lacks of a proper description in both figure legend and method section (What is the % of input lysate used? What was used as negative control? Is the IP performed on total, nuclear or chromatin extract?). In addition, a WB showing SALL4 protein levels in SALL4 IP is necessary to evaluate IP efficiency. What about the other HDACs?

ii) In Figure 4 and Figure 6E, the authors make large use of GO and GSEA analyses to correlate RNA-seq and H3K27ac ChIP-seq data upon SALL4 knockdown. However, this evidence does not prove that SALL4 is directly regulating those genes. Since SALL4 DNA binding motif is well characterized (see Tatetsu et al., Gene 2016) the authors should use TF enrichment analysis tools (e.g. ChEA) to evaluate if SALL4 is enriched at the set of genes differentially expressed/with a differential H3K27ac peak upon SALL4 knockdown.

iii) In order to claim a direct role for SALL4 in the epigenetic control of melanoma invasiveness, the authors should perform SALL4 ChIP-seq in melanoma cells. A number of publications show efficient SALL4 ChIP-seq experiments and the ChIP-grade antibody used in this study should allow the authors to perform this key experiment.

3. In Figure 4C, genes are selected that show evidence of an invasive signature, but it appears that genes are selected that fit the hypothesis by listing genes of interest. There are many other pathways coming up in 4B that might be important such as immune response that could contribute to metastasis. Also, in Figure 6d, the overlapping genes are immediately assumed to be invasive, but very few genes are listed. What are the unbiased GO terms for the overlapping genes (107 up and 91 down) (i.e. not just comparing to phenotype-switching signatures). Moreover, protein level analyses are not shown for key invasive genes such as MITF and Wnt5A, etc. These points should be addressed.

4. More conceptually, Sall4 is an embryonic stem cell factor that can be re-expressed by tumor cells, and the authors draw a parallel between SALL4 and other neural crest (NC) TFs, as well as the neurotrophin receptor CD271/NGFR of NC and, when re-expressed in melanoma cells,

promotes invasiveness, as reported in a publication from the same lab (Restivo et al., 2017). This is in line with the high migratory capacity of NC cell, supporting the notion that the acquisition of stem cell properties in tumor cells is associated with a de-differentiation process and EMT. However, the authors propose an opposite function for SALL4. When re-expressed in hyperplastic melanocyte, this TF favors primary melanoma growth and inhibits migration and invasion. Thus, it does not parallel the function of NC genes, but rather a more ESC phenotype. This apparent incongruence is never raised in the manuscript. If not experimentally, the authors need to address this point with at least a thorough justification in the discussion.

Minor points:

1. [LINE 503-504] The technical details of the RNA-seq experiment comparing WT and hyperplastic melanocytes are not present in the method section and the authors only refer to the original paper (Varum et al., 2019 from the same lab). To favor experimental reproducibility, it would be good practice to include experimental details instead of referring to previous publications. In particular in this case, given the fact that the original paper does not provide any description of RNA library preparation or data analysis.
2. [LINE 111-113] In figure 2c,d the authors show how Sall4 KO impairs tumor formation, although heterozygous KO animals do not show any phenotype. The quantification of Sall4 levels in Sall4^{+/-} mice would help understanding this phenotype (e.g. is Sall4 expression compensated by the remaining allele?).
3. [LINE 241] H4K27ac is a typo.
4. Fig S1, no quantification of lung metastasis is shown, although stated in the figure legend. Please add.

Addressing Editor and Reviewers Comments for Nat Commun from Aug 2020

REVIEWER COMMENTS

Reviewer #1 (Remarks to the Author):

Diener et al. study the role of the Stem cell factor SALL4 on melanomagenesis and melanoma metastatic spread. The authors combine genetic mouse model, transcriptomic analysis and in vitro assays in human melanoma cells to conclude that SALL4 expression regulates melanoma initiation and cell proliferation while its loss induces an undifferentiated, pro-metastatic melanoma phenotype similar to the previously published invasive $AXL^{high}/MITF^{low}$ phenotype. Furthermore, the authors postulate that this phenotype change is mediated by epigenetic mechanisms that involve HDAC mediated histone modifications. While there are some points to be raised, in general the data is clear and solid and the title reflects the conclusions the Diener et al. take from their results.

We thank the reviewer for her/his overall positive judgement and enthusiasm for our research, highly appreciate her/his inputs and are happy to present in this revised version of our study new experiments and analyses addressing her/his specific questions.

1. Figure 1b. Validation of SALL4 mRNA by qRT-PCR expression should be presented.

To validate our findings in an independent experiment, we have isolated GFP^+ or $tdTomato^+$ murine melanocytes from murine skin with different genotypes and have included again wild-type animals and $Tyr::Nras^{Q61K} Cdkn2a^{-/-}$ mice as in Figure 1a-b. To get enough cells from the murine skin for downstream analysis, we have pooled sorted cells from 6-10 animals per genotype. The revised Supplementary Figure S3b now shows relative mRNA levels of *Sall4* – assessed by means of RT-PCR – in newly isolated wild-type and hyperplastic melanocytes (together with hyperplastic melanocytes with heterozygous or homozygous *Sall4* loss – in response to your point #4). These data confirm the increase (more than 10 fold) of *Sall4* expression in hyperplastic melanocytes ($Tyr::Nras^{Q61K} Cdkn2a^{-/-}$) compared to wild-type murine melanocytes.

2. Figure 1c. If possible images with better resolution should be presented.

We now present these images (with slightly changed image frames) in higher resolution. Also, for clarification, arrowheads now point towards *Sall4* negative wild-type melanocytes in hair follicles and arrows point towards *Sall4* positive hyperplastic interfollicular melanocytes present in $Tyr::Nras^{Q61K} Cdkn2a^{-/-}$ mice.

3. Figure 2 shows that heterozygous KO of SALL4 doesn't affect the number of tumors. What's the proliferation/mitotic rate in these tumors compared to SALL4 "wt" animals?

This is an important point and we have now addressed this issue in the revised manuscript. The novel Figure 2e shows the quantification of Sox10 (melanoma marker) and Ki67 in skin tumors of control ($Tyr::Nras^{Q61K} Cdkn2a^{-/-} Sall4^{wt/wt}$) vs. heterozygous *Sall4* cko ($Tyr::Nras^{Q61K} Cdkn2a^{-/-} Sall4^{lx/wt}$) mice (Figure 2e and Supplementary Figure S3c). These results show that primary tumors with heterozygous *Sall4* loss have a significantly impaired proliferation rate compared to primary tumors of $Sall4^{wt/wt}$

animals. However, although $Sall4^{lx/wt}$ tumors were characterized by a lower positivity for Ki67, the overall number of tumors didn't significantly change, suggesting that melanoma initiation was not affected (Figure 2d; as shown in the first version of this manuscript).

4. Figure 2d. What is the expression of SALL4 in +/+, +/- and -/- tumors at the mRNA level?

Since $Sall4^{-/-}$ cko ($Tyr::Nras^{Q61K}$ $Cdkn2a^{-/-}$ $Tyr::Cre^{ERT2}$ $Sall4^{lx/lx}$) animals do not develop primary tumors, we decided to address this question by isolating murine melanocytes from neonatal skin of the three different genotypes directly after recombination (tamoxifen was applied to the breast-feeding mother one day after birth and pups were euthanized 8 days after birth). By FACS we sorted melanocytes expressing either GFP or tdTomato (LSL-R26R-GFP or LSL-R26R-tdTomato) from $SALL4^{+/+}$ ($Tyr::Nras^{Q61K}$ $Cdkn2a^{-/-}$ $Tyr::Cre^{ERT2}$ $Sall4^{wt/wt}$), heterozygous $Sall4^{+/-}$ cko ($Tyr::Nras^{Q61K}$ $Cdkn2a^{-/-}$ $Tyr::Cre^{ERT2}$ $Sall4^{lx/wt}$), homozygous $Sall4^{-/-}$ cko ($Tyr::Nras^{Q61K}$ $Cdkn2a^{-/-}$ $Tyr::Cre^{ERT2}$ $Sall4^{lx/lx}$) and wild-type animals. Note that 6-10 animals of the same genotype had to be pooled to get enough cells per sample for downstream application. We then assessed *Sall4* expression levels by RT-PCR. The novel Supplementary Figure S3b shows that hyperplastic melanocytes ($Tyr::Nras^{Q61K}$ $Cdkn2a^{-/-}$ $Tyr::Cre^{ERT2}$ $Sall4^{wt/wt}$) have more than 10 fold increased *Sall4* expression compared to wild-type melanocytes. Moreover, upon recombination, *Sall4* expression levels decrease to approximately 50 % in heterozygous *Sall4* cko melanocytes, and its expression decreases even further in homozygous *Sall4* cko melanocytes to levels comparable to wild-type melanocytes.

5. Figure 3a-f: Partial knock-down of human SALL4 by siRNA clearly affects proliferation, would this contradict the results obtained in Figure 2, regarding heterozygous SALL4 KO? This point should be addressed (see point 3).

This is an important point that required further clarification. We now show in Figure 2e that while *Sall4* heterozygous cko tumors can be formed (Figure 2c, d), they are characterized by decreased proliferation rates (compared to *Sall4* wild-type tumors). These data corroborate our findings obtained by *SALL4* knock down in human cell lines in vitro (Figure 3d-f), which also leads to significantly decreased cell proliferation.

6. Since the mouse model assesses melanoma formation in a mutNRAS background, I think the authors could have assessed in several mutRAS cell lines (WM1361, WM1366, SBCL2...). At present the authors only used a "pure" NRAs cell line (M01) to assess proliferation, migration and invasion. Still it would be interesting to assess if SALL4 regulates proliferation and invasion independently of the driver mutation (BRAF v NRAS) regulating melanoma cell biology.

We agree with the reviewer that it was important to broaden our analysis and we have now added additional cell lines to our experiments covering a bigger array of different mutational backgrounds: $NRAS^{Q61K}$, $BRAF^{V600E}$, $NRAS^{Q61K}$ $BRAF^{V600E}$ double-mutant, $NRAS^{Q61R}$ $PTEN^{+/-}$ (as suggested by the reviewer) and one cell line with unknown mutational status. The impaired proliferation upon *SALL4* knock down (Figure 3d-f and Supplementary Figure S5a-b; proliferation assessed in two different cell lines with $BRAF^{V600E}$, one with $NRAS^{Q61K}$, one with $NRAS^{Q61R}$ $PTEN^{+/-}$ and one with unknown mutational status), as well as the differential expression of invasiveness genes (Figure 4e; invasiveness gene expression assessed in one cell line with $NRAS^{Q61K}$, two with $BRAF^{V600E}$, one with $NRAS^{Q61K}$ $BRAF^{V600E}$, and one with unknown mutational background) suggest that the observed *SALL4*-mediated phenotypes are independent of the main driver mutations (at least *BRAF* and *NRAS* g.o.f. mutations).

7. Does SALL4 regulate an undifferentiated phenotype in mutBRAF human melanoma cells?

As mentioned above, our data suggest that impairment of SALL4 (by siRNA) leads to reduced proliferation (Figure 3d-f, Supplementary Figure S5a-b), increased expression of invasiveness genes (Figure 4b, d, e) and decreased expression of melanocyte differentiation genes (Figure 4c, d, e) independently of the mutational background of the cell lines (Figure 4e). For instance, in the BRAF^{V600E}-mutated human melanoma cell lines MM150548 and MM150536, SALL4 knock down leads to downregulation of MLANA or DCT respectively (Figure 4e), two crucial genes for melanocyte pigmentation, which together with the upregulation of stemness-associated factors like NGFR, JUN, CDH2 and others (Figure 4e) can be understood as an ‘undifferentiated phenotype’. Importantly, we now show that SALL4 loss-mediated phenotypes are not dependent on the driver mutations, but that they are observed also in melanoma cell lines wildtype for BRAF.

If that is the case it would open an interesting hypothesis regarding the role of SALL4 in the regulation of mutBRAF melanoma cell response to MAPK inhibitors. Is SALL4 differentially expressed in melanoma tumors treated with BRAF/MEK inhibitors? I would advise the authors to perform such studies to strengthen the relevance of the manuscript.

To address whether SALL4 expression changes in response to BRAF inhibition, we analyzed existing RNA sequencing data of a big set of human melanoma cells from the biobank of the University Hospital Zurich, Switzerland. Specifically, human tumor-derived cell lines for which RNA seq had been performed, were treated in vitro with Encorefenib for 72 hrs (concentrations based on IC50 values from Resazurin cell viability assays) and categorized into BRAFi ‘resistant’ versus ‘sensitive’ (Figure below, left panel). We didn’t find any difference in SALL4 expression levels in those cell lines that are either BRAFi resistant or sensitive. We further sub-categorized the same cells based on their clinical history: whether the patients from which those cells derive had been treated with BRAFi (‘BRAFi treated’) or not (‘BRAFi naïve’) (Figure below, right panel). However, there was again no statistically significant difference in SALL4 expression between BRAFi naïve and treated patient-derived cell lines.

Since these results do not reveal differential expression of SALL4 in response to BRAF inhibition, evidence for a putative role of SALL4 in drug sensitivity and/or resistance formation is currently lacking, although we can formally not rule out a potential functional implication of SALL4 in drug response independently of changes in its expression. However, we feel that further analysis of this point would go beyond the scope of the current study and propose not to include these data in the manuscript in order to keep its flow and focus. We strongly feel that, even without a link to targeted therapy

responses, our study on a stem cell factor regulating cellular plasticity in melanoma by epigenetic control mechanisms is of high scientific interest and potential clinical relevance.

8. Page 7, lines 135-141. Apparently, tumors were analyzed “within 6 days after grafting”. Yet these tumors reached a volume of 0.05cm³/5000mm³. Is there an error in the text or in the graph scale in Figure 3h?

We have calculated tumor graft volumes in the range of 17-64 mm³ (= 0.017-0.064 cm³) for M010817 and 20-81 mm³ (= 0.020-0.081 cm³) for M150548 xenografts. Both cell lines are highly proliferative and, 6 days after injection of approx. 1'000'000 cells, we were already able to measure tumors with such a volume.

9. Figure 4C. Validation of mRNA levels from relevant genes should be included.

RT-PCR-based validation of invasiveness/differentiation gene expression after SALL4 KD had already been presented in the original Figure 4. The validation has now been extended to even more cell lines (5 in total) treated with two different siRNAs (novel Figure 4e).

10. Figure 5a. A blot showing the levels of immunoprecipitated SALL4 needs to be presented in the main figures. The same applied to Suppl. Figure S3a.

We agree that this was an important control to add. We have now repeated the Co-IP experiment, and we show the efficiency of the pulldown of the SALL4 protein itself (Figure 5a and Supplementary Figure S7).

11. Figure 5f. What are the levels of MITF in these tumors?

We have now assessed the expression of the melanocyte differentiation genes MITF, MLANA and DCT in the xenograft tumor (Supplementary Figure S12c) and found that while the differential expression of those genes is not significant, there is a trend for their downregulation in Mocetinostat (HDAC class I inhibitor)-treated grafts compared to vehicle-treated grafts (Supplementary data S6b).

Reviewer #2 (Remarks to the Author)

The manuscript by Diener, Baggiolini et al., explores the potential role of the stem cell factor SALL4 on the phenomenon of phenotype switching in melanoma. The authors found that Sall4 is elevated in hyperplastic murine melanocytes using a well-characterized GEMM and that its loss reduces primary tumor formation while increasing metastatic burden. The authors then study SALL4 function in human melanoma cells, and propose that this transcription factor (TF) regulates melanoma phenotype switching by suppressing an invasive gene signature through histone deacetylase recruitment. The manuscript presents robust phenotypic data and the generation of a Sall4 KO mouse melanoma model is an important achievement that might be beneficial for further studies. However, the epigenetic mechanism by which SALL4 regulates gene expression has only been partially addressed and the SALL4-HDAC2 mechanistic link is not supported by solid experimental data. Other issues include cherry-picking of gene lists without providing a full view of the data and the lack of Sall4 ChIP-seq.

Therefore, while an interesting study, several important shortcomings must be addressed as described below.

We thank the reviewer for her/his positive feedback on our mouse model study, her/his critical reading of our manuscript and some essential constructive criticism on the postulated molecular function of SALL4 in regulating invasiveness. We are happy to present with the revised manuscript novel experiments and analyses addressing her/his points of concern. Amongst others, we have now generated more extensive Supplementary Data Files with all obtained significant results from various sequencing results together with the complete list of Gene Ontology analyses to circumvent the impression of biased 'cherry-picking' of genes of interest.

More importantly, we have now performed CUT&RUN analyses for SALL4 to experimentally identify direct targets of SALL4. Moreover, to strengthen our hypothesis of a SALL4-HDAC2 shared molecular function in regulating melanoma invasiveness, we have also performed CUT&RUN for HDAC2 and analyzed the peaks shared between the two factors (novel Figures 5 and 6). Interestingly, our analysis revealed a big set of shared SALL4-HDAC2 peaks supporting the idea of a major joint function of SALL4 and HDAC2 in melanoma. In particular, we found a big set of shared SALL4-HDAC2 CUT&RUN peaks annotated to prominent invasiveness genes that we had found upregulated after SALL4 knock down as well as after HDAC inhibition. We are convinced that these novel results substantially improve the quality and impact of our study, especially with respect to the mechanism of how SALL4 molecularly regulates melanoma cell invasion.

Please find our specific replies to your questions below.

Major points:

1. Sall4 depletion in the mouse models promotes metastasis and the mouse data is compelling. The authors then examine Sall4 function in human melanoma cells. In Figure 3 the authors perform siRNA-mediated knockdown in human melanoma cells and validate their findings from the mouse model with multiple phenotypic assays. However, the efficiency of SALL4 knockdown by mRNA expression is poor in all three melanoma lines, in particular in M12 cells where it barely reaches 30% reduction (Figure 3a). In contrast, Western blot analysis of SALL4 knockdown in M01 cells appears to totally deplete SALL4 (Supplementary Figure S2A). This discrepancy needs to be clarified. SALL4 protein levels upon knockdown should be assessed in all three melanoma cell lines and presented in the main figure to better correlate SALL4 knockdown levels with the different degrees of phenotypic changes observed.

We thank the reviewer for pointing this out. We have now added western blots of SALL4 protein after siRNA-mediated knock downs (Figure 3a-c), which show that on protein levels, the knock down efficiency is quite strong (ca. 80-100% in M010817; ca. 50-100% in M070302 and ca. 40% in M150548). Strikingly, especially for M010817 and M070302 with almost complete SALL4 loss on protein levels (Figure 3a, b), RT-PCR analysis revealed considerably weaker knock down efficiencies on mRNA levels (Figure 3 a, b lower panels), even though protein and RNA samples were derived from the exact same samples (split in half). To exclude the possibility that this is due to unspecific primers, we analyzed the same sample (M010817 treated with siRNA#1 from Figure 3a) with other sets of published SALL4 primers (Kobayashi et al. (2011) *International Journal of Oncology*, 38(4), 933–939; Lin et al. (2015) *Scientific Reports*, 5, 1–12) (Figure below for the reviewers' discretion). Again, also when using those

previously published primers, SALL4 mRNA knock down seemed moderate even though protein levels were strongly reduced (Figure 3a top panel).

We wondered whether the knock down might specifically decrease the SALL4 isoform A (the main SALL4 isoform, 160 kDA, that is presented by the western blots in Figure 3a-c), but not affect the expression levels of the other isoforms (B and C). Therefore we analyzed the same M010817 sample by RT-PCR with published primers for the specific isoforms (Yong et al. (2013) *New England Journal of Medicine*, 368(24), 2266–2276). Similarly using isoform specific primers, the mRNA levels of neither isoform was decreased more than approx. 40% (Figure below).

Overall, we observed that while SALL4 knock down efficiency is moderate on mRNA levels, its effect on protein levels is striking, suggesting a stronger impact on mRNA translation and/or protein stability.

In addition, Figure S2A shows how SALL4 levels are much lower in melanoma cells than in the control embryonal carcinoma cell line NTERA2D1. What is the expression level of this protein in normal human melanocytes? This data would help validating the theory of SALL4 specific re-expression in melanoma tumors.

To address this point, we refer to a past study (Reemann et al., 2014; lines 98ff), which showed by RNA sequencing of adult human melanocytes that SALL4 is not expressed. We have now validated this finding by immunohistochemistry (IHC) on human healthy skin and human primary melanoma. Strikingly, we couldn't detect any SALL4 expression in healthy human melanocytes, while we observed its expression in the human primary tumors (Supplementary Figure 2).

The authors reach Sall4 through RNA-seq data presented in Figure 1, but they don't provide the full picture and only present top 20 genes. They go on to state that Sall4 is the top TF, but don't provide any evidence of other TFs.

We agree with the reviewer that one should avoid 'cherry picking' of factors of interest and apologize for not having made clear in the original version of our manuscript why we focused on Sall4. We have now generated a novel Supplementary Data file (Supplementary Data 1), which includes the entire list of differentially expressed genes comparing hyperplastic to wild-type melanocytes. In the complete list, we can find additional TFs, but Sall4 remains the TF with the biggest fold change in hyperplastic melanocytes and the only TF in the top 20 upregulated genes from Supplementary Data 1 (Figure1b).

Moreover, they only examine hyperplastic state of melanocytes. How do Sall4 levels compare in the melanomas that eventually develop in these mice? If its expression is elevated in hyperplastic state, does this correlate with proliferation of these cells?

We have now included a immunohistochemical staining of Sall4 in hyperplastic skin as well as a primary tumor (from the same animal) of our GEMM (Supplementary Figure S1a-b). The data show that Sall4 protein is vastly expressed in the tumors, while in the hyperplastic skin Sall4 is heterogeneously expressed and we could detect both positive and negative interfollicular melanocytes (a characteristic of the hyperplasia state in this mouse melanoma model).

Indeed, as the reviewer suggests, Sall4 levels correlate with proliferation as we now show in Figure 2e. Measuring proliferation rates in Sall4 wild-type and Sall4^{+/-} cko tumors revealed that Sall4^{+/-} cko tumors have a significantly decreased (ca. 50%) proliferation rate compared to Sall4 wild-type tumors (please see also our response to Reviewer 1, point #3).

2. The change in chromatin state caused by SALL4 depletion is key to the mechanistic value of this study. However, the mechanism by which SALL4 recruits HDAC2 on specific gene targets is not clear. In fact, the author's conclusions about SALL4's mechanism of action are based on the indirect evidence that histone acetylation is impaired in SALL4 knockdown cells. This is a substantial concern that needs to be address with different lines of experimental data as suggested here:

i) The Co-Immunoprecipitation experiment demonstrating the interaction between SALL4 and HDAC2 is a crucial point in the manuscript (Figure 5a). However, this assay lacks of a proper description in both figure legend and method section (What is the % of input lysate used? What was used as negative control? Is the IP performed on total, nuclear or chromatin extract?). In addition, a WB showing SALL4 protein levels in SALL4 IP is necessary to evaluate IP efficiency. What about the other HDACs?

We agree with the reviewer that the link between SALL4 and HDAC2 required further investigation and we appreciate the comments made, which considerably strengthened the manuscript.

We have now added a precise description of the method in the Material & Methods section to explain in details how the Co-IP experiments were performed (page 31, line 719ff). We have also included a western blot showing SALL4 protein levels after pulldown (Figure5a and Supplementary Figure S7). Finally, we have expanded the analysis to different HDACs, but we couldn't detect any enrichment for HDAC1, HDAC4 or HDAC6 (Supplementary Figure S7).

To show whether SALL4 and HDAC2 share common targets and bind to the same peaks, we performed a CUT&RUN sequencing experiment for both SALL4 and HDAC2 (with two different antibodies for each factor) and analyzed loci that presented with significant peaks for at least 3 of 4 antibodies against SALL4 and HDAC2. Strikingly, we observed that 28% of all SALL4 peaks had peaks with minimum 3 of 4 antibodies (Figure 5c), which further corroborates that SALL4-HDAC2 co-functioning is an important part of the SALL4-mediated regulatory mechanism in melanoma cells (please also see point below 2ii).

ii) In Figure 4 and Figure 6E, the authors make large use of GO and GSEA analyses to correlate RNA-seq and H3K27ac ChIP-seq data upon SALL4 knockdown. However, this evidence does not prove that SALL4 is directly regulating those genes. Since SALL4 DNA binding motif is well characterized (see Tatetsu et al., Gene 2016) the authors should use TF enrichment analysis tools (e.g. ChEA) to evaluate if SALL4 is enriched at the set of genes differentially expressed/with a differential H3K27ac peak upon SALL4 knockdown.

iii) In order to claim a direct role for SALL4 in the epigenetic control of melanoma invasiveness, the authors should perform SALL4 ChIP-seq in melanoma cells. A number of publications show efficient SALL4 ChIP-seq experiments and the ChIP-grade antibody used in this study should allow the authors to perform this key experiment.

We agree with the reviewer that the link between SALL4 and HDAC2 required further investigation and we appreciate the comments made, which helped to considerably strengthen the manuscript. To address this important issue, we have now performed CUT&RUN experiments for SALL4 as well as for HDAC2 to discover the targets that are bound by both factors (Figures 5 and 6 and 7).

In Supplementary Data 3, we now provide the full lists of SALL4 and HDAC2 CUT&RUN peaks. Importantly, we have analyzed the SALL4-HDAC2 shared peaks (Supplementary Data 4) related to genes whose expression was altered in the RNA-seq data (Figure 6). Overall, this analysis shows that a set of melanoma invasiveness genes are direct targets of both SALL4 and HDAC2 and that these genes are upregulated upon SALL4 loss, e.g. CDH2, FN1, TGFBR2, VEGFR1, PDGFC, ITGA6 among others. We also show that after SALL4 depletion there is an increase in the transcription activating mark H3K27ac in melanoma invasiveness genes, such as FN1, VEGFR1, PDGFC, ITGA6, NGFR, AXL and others (Figure 7). Finally, many SALL4-HDAC2 targets that are upregulated upon SALL4 knock down, are also upregulated upon HDAC inhibitor treatment (Figure 6) and, vice versa, their upregulation upon SALL4 knock down can be rescued by co-treatment with an inhibitor of histone acetyl transferases (Supplementary Figure S15). These data support the hypothesis that SALL4 and HDAC2 negatively regulate a set of invasion genes, which get activated upon SALL4 or HDAC inhibition, at least partially through epigenetic rewiring.

In regard to the SALL4 DNA binding motifs, we have also assessed SALL4, HDAC2 and SALL4-HDAC2 shared motifs (Supplementary Figures S8-S10) and compared them with recent studies that have also performed SALL4 C&R (on different cell types) (Kong et al., 2021 and Pantier et al., 2021). The AT-rich motif suggested to be a hallmark for SALL4 DNA-binding motifs in these studies was detected in 38% of our SALL4 peaks. However, since such a motif was not centrally localized within the DNA fragments and since the DNA motif itself is not particularly complex (AWTATKR), the biological relevance of this motif in melanoma remains to be addressed.

3. In Figure 4C, genes are selected that show evidence of an invasive signature, but it appears that genes are selected that fit the hypothesis by listing genes of interest. There are many other pathways coming up in 4B that might be important such as immune response that could contribute to metastasis.

To be more comprehensive, we now present the differentially expressed genes as top 12 enriched Networks in MetaCore and added the top 10 changed genes per pathway (Figure 4b,c and all other Figures with pathway enrichment analyses). These data confirm that many of the top enriched Process Network of UPREGULATED genes indeed enrich in pathways associated with invasiveness, as now shown by the list of specific differentially expressed genes representing them. Indeed, other pathways such as the immune-related pathways are of big potential interest as well and we hope to be able to address this interesting aspect of SALL4-mediated regulation of an immune response in future studies.

Also, in Figure 6d, the overlapping genes are immediately assumed to be invasive, but very few genes are listed. What are the unbiased GO terms for the overlapping genes (107 up and 91 down) (i.e. not just comparing to phenotype-switching signatures).

Again, the reviewer raises an important point. Therefore, as mentioned above, we have now included the complete GO terms and listed either all differentially expressed genes per pathway (Figures 6b, c and 7I and Supplementary Figures S11b and S13b) or the top 10 most changed genes in case of bigger datasets (as in Figure 4b, c). The full GO analyses show that the direct SALL4-HDAC2 targets that are also upregulated in the RNA-seq strongly enrich in cell adhesion, cytoskeleton, angiogenesis and EMT

pathways, i.e. all pathways associated with invasiveness and metastasis formation (Figure 6b). Likewise, target genes related to invasiveness also show an increase in H3K27ac peaks around the TSS (Figure 7h, i). As mentioned, the reader finds now the entire lists of GO terms (p-value threshold < 0.05) for all these analyses in the Supplementary Data Files as separate xls sheet.

Moreover, protein level analyses are not shown for key invasive genes such as MITF and Wnt5A, etc. These points should be addressed.

We now show elevated protein levels of key invasiveness genes like NGFR and AXL after SALL4 knock down (Supplementary Figure S6b) and also show the upregulation of invasiveness genes on the protein level by means of immunocytochemistry after SALL4 knock down on cells in vitro (Supplementary Figure S6a). Reduction on the protein level of melanocytes differentiation genes such as MLANA upon SALL4 KD can further be found in Supplementary Figure S15b, c.

4. More conceptually, Sall4 is an embryonic stem cell factor that can be re-expressed by tumor cells, and the authors draw a parallel between SALL4 and other neural crest (NC) TFs, as well as the neurotrophin receptor CD271/NGFR of NC and, when re-expressed in melanoma cells, promotes invasiveness, as reported in a publication from the same lab (Restivo et al., 2017). This is in line with the high migratory capacity of NC cell, supporting the notion that the acquisition of stem cell properties in tumor cells is associated with a de-differentiation process and EMT. However, the authors propose an opposite function for SALL4. When re-expressed in hyperplastic melanocyte, this TF favors primary melanoma growth and inhibits migration and invasion. Thus, it does not parallel the function of NC genes, but rather a more ESC phenotype. This apparent incongruence is never raised in the manuscript. If not experimentally, the authors need to address this point with at least a thorough justification in the discussion.

Thank you for having raised this intriguing issue. There is increasing evidence that neural crest stem cell (NCSC) factors are re-expressed in melanoma and have been implicated in tumor initiation and progression (see also our recent review, Diener and Sommer, 2020). However, unlike melanoma cells undergoing phenotype switching from proliferative to invasive states, NCSCs during embryonic development continue to proliferate as they migrate. Intriguingly, while some NCSC factors, such as SOX10 and YY1 (both required for proliferation and survival of NCSCs), are activated upon melanoma formation and required for melanoma growth, other NCSC-associated factors, such as CD271/NGFR/p75^{NTR}, PAX3, and FOXD3 promote melanoma cell invasiveness and metastasis formation. Therefore, we suggest that the embryonic program active in NCSCs segregates in melanoma to regulate distinct aspects of phenotype switching, proliferation vs. invasion. SALL4 falls in the same category as SOX10 and YY1 and supports melanoma cell proliferation. Importantly, however, SALL4 depletion in melanoma leads to increased invasiveness and micrometastasis formation. This is reminiscent of melanoma cells with reduced SOX10 levels, which also display increased invasiveness. Thus, in melanoma, part of a NCSC program associated with proliferation appears to counteract another part of the NCSC program, which regulates invasiveness, although the molecular mechanisms underlying these antagonistic roles of NCSC-associated factors in melanoma biology remain to be elucidated. Following this reviewer's suggestion, we now discuss this interesting point in an extended Discussion (Page 19, lines 430).

Minor points:

1. [LINE 503-504] The technical details of the RNA-seq experiment comparing WT and hyperplastic melanocytes are not present in the method section and the authors only refer to the original paper (Varum et al., 2019 from the same lab). To favor experimental reproducibility, it would be good practice to include experimental details instead of referring to previous publications. In particular in this case, given the fact that the original paper does not provide any description of RNA library preparation or data analysis.

We have now included a detailed description of this experiment for the RNA sample preparation, reverse transcription, library prep, sequencing and the subsequent analysis. Please find all details in the Materials & Methods section on page 29, line 670ff.

2. [LINE 111-113] In figure 2c,d the authors show how Sall4 KO impairs tumor formation, although heterozygous KO animals do not show any phenotype. The quantification of Sall4 levels in Sall4^{+/-} mice would help understanding this phenotype (e.g. is Sall4 expression compensated by the remaining allele?).

We have now addressed this issue by two means (please see also our responses to Reviewer 1, points #3 and #4): First, we assessed Sall4 mRNA levels in Sall4 wild-type, heterozygous Sall4^{+/-} cko, and homozygous Sall4^{-/-} cko melanocytes. Our data presented in a novel Supplementary Figure S3b confirm that Sall4^{+/-} cko melanocytes express Sall4 at approx. 50% of the Sall4 levels found in wild-type melanocytes. This goes in line with our second novel analysis, showing that while Sall4^{+/-} cko animals do form primary tumors, they are significantly impaired in their proliferation rate (Figure 2e). We now discuss these data in the revised manuscript (Lines 122ff).

3. [LINE 241] H4K27ac is a typo.

This has been corrected. Thank you.

4. Fig S1, no quantification of lung metastasis is shown, although stated in the figure legend. Please add.

This figure (now Supplementary Figure S4a, b) only serves visualization purposes for the reader to understand how the endogenous GFP signal looks like before conversion into the black and white images presented in Figure 2f and Supplementary Figure S4b. The quantification of the lungs (in total 67 lungs, i.e. many more than visualized for illustration purposes in the main and supplementary figures) is presented in the main Figure 2h.

REVIEWER COMMENTS

Reviewer #1 (Remarks to the Author):

The authors have satisfactorily addressed my comments and the manuscript is now clearly improved.
I have no more comments.

Reviewer #2 (Remarks to the Author):

Diener, Baggiolini et al. have provided a substantial revision of their manuscript "Epigenetic Control of Melanoma Cell Invasiveness by the Stem Cell Factor Sall4", which now appears greatly improved. They have addressed key issues raised by both reviewers namely, genomic evidence supporting the direct role of SALL4 and HDAC2 in the coordinated epigenetic control of a set of invasive genes. Therefore, while some minor aspects of this work can be further improved as noted below, the overall conclusion of the manuscript is now well supported by the newly presented data and the paper is recommended for publication.

One area of concern involves how the new epigenomic data are presented in the updated Fig 5, as well as their integration with previous analyses in Fig 6-7, that would benefit a more rational presentation.

1. While the use of two different antibodies to analyze SALL4 and HDAC2 chromatin binding is appreciated, the decision to only select peaks shared in at least 3 out of 4 conditions is not clear. Here the authors are showing only the data that support their initial hypothesis (SALL4/HDAC2 cooperative chromatin binding), although that is just a fraction of the regions bound by these chromatin factors (only 28% of SALL4 sites are shared). The authors should present SALL4 and HDAC2 CUT&RUN regions as side-by-side read density heatmaps to visualize what is the percentage of shared peaks on the total identified regions. Panel 5b is not as informative. Furthermore, if the rationale for employing two antibodies for each protein is improving specificity, then the authors should select only peaks identified by both antibodies. Otherwise, the authors should choose one of the two antibodies for downstream analyses, in particular if one of them presents a reliable/expected signal as has been reported in the literature with confidence. This is particularly striking in the case of HDAC2 antibodies: ab#1 identifies 3,500 peaks, ab#2 almost 10,000. Is it possible that ab#1 simply did not work? Checking publicly available HDAC2 ChIP-seq/CUT&RUN data would be helpful in this context.

Also, the authors should adapt genomic snapshots in Fig 5e-i to be consistent with other figures in the paper, such as in Fig 7c-g (bar chart vs. line plot in IGV). Also, adding genomic coordinates along with a more comprehensive representation of gene TSS/exons/introns (as shown in IGV) instead of a stylized drawing, would be more appropriate here.

2. The integration of SALL4/HDAC2 CUT&RUN with RNA-seq and H3K27ac ChIP-seq is a key analysis in the paper, but is somehow missing. Although the authors have demonstrated that SALL4 is responsible for HDAC2-mediated downregulation of some invasive genes, the ChIP-seqs have not been integrated to put the whole story together.

Minor points:

- While the authors show some data in the Supp of Sall4 staining of human melanoma, it would be important to search publically available data sets, such as TCGA to see if SALL4 expression correlates with metastasis.
- The heatmap illustrating RNA-seq data in Fig 4a could be better represented by volcano plot highlighting significant genes.
- The term RT-PCR used in the result section is misleading when referring to qRT-PCR, as correctly stated in the method paragraph. Please correct.

Addressing Reviewers Comments for Nat Commun from May 2021 (Revision II)

NCOMMS-20-25425A

REVIEWER COMMENTS:

Reviewer #1 (Remarks to the Author):

The authors have satisfactorily addressed my comments and the manuscript is now clearly improved.

I have no more comments.

We thank the reviewer for his/her positive feedback and are happy that we satisfactorily met his/her requirements for a revised version of our scientific study.

Reviewer #2 (Remarks to the Author):

Diener, Baggiolini et al. have provided a substantial revision of their manuscript "Epigenetic Control of Melanoma Cell Invasiveness by the Stem Cell Factor Sall4", which now appears greatly improved. They have addressed key issues raised by both reviewers namely, genomic evidence supporting the direct role of SALL4 and HDAC2 in the coordinated epigenetic control of a set of invasive genes. Therefore, while some minor aspects of this work can be further improved as noted below, the overall conclusion of the manuscript is now well supported by the newly presented data and the paper is recommended for publication.

We appreciate that the reviewer finds our revised study substantially improved and agree that the novel data supporting a direct role of SALL4 and HDAC2 on melanoma invasiveness gene expression added great value to the molecular mechanism underlying our work.

One area of concern involves how the new epigenomic data are presented in the updated Fig 5, as well as their integration with previous analyses in Fig 6-7, that would benefit a more rational presentation.

1. While the use of two different antibodies to analyze SALL4 and HDAC2 chromatin binding is appreciated, the decision to only select peaks shared in at least 3 out of 4 conditions is not clear.

We appreciate that the reviewer pointed out that the analysis of the CUT&RUN peaks might be presented in an unclear fashion. Our intention was to use commonly used antibodies for SALL4 as well as HDAC2 that had also been used for CUT&RUN or comparable (i.e. ChIP seq) experiments before. Namely those were the antibodies SALL4 Ab#1 (ab29112) and HDAC2 Ab#1 (ab12169). However, to strengthen our analysis we wanted to add an additional antibody per factor resulting in 2 different antibodies for each SALL4 and HDAC2.

We felt that while analyzing peaks that were shared between all 4 antibodies would obviously give the most stringent set of genes bound by SALL4 and HDAC2, we would prefer to include also genes that showed peaks for 3 of 4 antibodies to minimize ruling out target genes as false negatives due to different reasons. First, the less frequently used antibodies have never before or only little been validated in ChIP or CUT&RUN sequencing experiments, hence their specificity for our purpose has little been investigated before. Second, a known problem in the field is that absolute concordance of identified peaks across ChIP-seq replicates is actually one of the major challenges - even when using the same

antibody -, which led to the widely used proposed approach of >50% of concordance across samples (see also 'Leveraging biological replicates to improve analysis in ChIP-seq experiments' by Yang et al., 2014, *Comput Struct Biotechnol J*, DOI: [10.5936/csbj.201401002](https://doi.org/10.5936/csbj.201401002)). This led us to the choice of using different antibodies – rather than sampling the same antibody twice –, which actually strengthens the validity of the peaks found for SALL4 and HDAC2. Third, the usage of two different antibodies for the peak identification of the same factor not only increases specificity, but potentially allows the discovery of new peaks that might be lost if specific epitopes are masked in some protein complexes at specific locations, which might only be detectable by one or the other antibody. Last but not least, the specificity of our CUT&RUN peaks is provided by the controls which allow removal of artifact peaks or sequencing biases, and by the stringent peak calling algorithms employed.

Here the authors are showing only the data that support their initial hypothesis (SALL4/HDAC2 cooperative chromatin binding), although that is just a fraction of the regions bound by these chromatin factors (only 28% of SALL4 sites are shared).

Indeed, as the reviewer states, we are focusing here on the genes bound by both SALL4 and HDAC2, however in the Supplementary Data we also present all the peaks for each single antibody so the reader has full access to the unique peaks as well. To improve the transparency of our obtained data, we have now included an extra Supplementary Data file (Supplementary Data 6) listing the SALL4 unique peaks and also HDAC2 unique peaks. To make a statement about a putative exclusive function of SALL4 independent of HDAC2 and vice versa (as we assume the reviewer implies with his/her question), we have further screened for genes that only had SALL4 (or HDAC2) peaks annotated to them but no shared peaks at all (Supplementary Data 6). With those 'exclusively SALL4 bound' or 'exclusively HDAC2 bound' genes we further performed pathway enrichment in MetaCore. According to this analysis, SALL4 exclusive targets enrich in neurogenesis and neurophysiology-related biological processes, while HDAC2 exclusive targets enrich in various different pathways related to immune cell adhesion, developmental pathways, cell cycle, apoptosis and others (Supplementary Data 6).

We now briefly discuss this point in the revised manuscript (lanes 301ff) and have listed the unique CUT&RUN peaks for SALL4, as well as HDAC2 plus the genes with exclusively one peak type (SALL4 only or HDAC2 only) in the novel Supplementary Data 6.

Nevertheless, since the purpose of this experiment - namely the CUT&RUN analysis of both, SALL4 and HDAC2 - was to find *shared* target genes, we believe that for the story flow of our study it is correct to specifically look at, and highlight, the C&R peaks shared between both factors. However, with the new Supplementary Data 6, the reader now also receives a more in-depth analysis of the genes exclusively bound by SALL4 or HDAC2 alone.

Importantly, and to catch up on the reviewer's concern, we believe that for a transcription factor (like SALL4) a whole third (as the reviewer mentions) of peaks shared with an epigenetic modifier (like HDAC2) actually speaks in favor of the interaction of the two being a major part of SALL4's function in melanoma. In fact, that an overlap between SALL4 and HDAC2 binding is detected is a remarkable fact per se: once artifact peaks are excluded by negative controls and in silico trimming of repetitive regions and PCR/seq biases, the resulting peaks are statistically very improbable events. Hence, any peak overlap is compelling evidence that SALL4 and HDAC2 function in the same protein complex at the identified genomic locations. Also, the 3 of 4 approach does not undermine this powerful observation but is an analytical choice and similar overlaps (quantitatively) would be measured even if considering only one antibody per protein.

The authors should present SALL4 and HDAC2 CUT&RUN regions as side-by-side read density heatmaps to visualize what is the percentage of shared peaks on the total identified regions. Panel 5b is not as informative.

We thank the reviewer for this suggestion and agree that a read heatmap is more informative than the selected tracks in our Figure 5b. We have therefore generated the read density heatmaps for the antibodies used (including the negative control), which can now be seen in Figure 5b, that show the centered peaks (within 10kb) that were identified with at least 3 of the 4 SALL4/HDAC2 antibodies. As one can appreciate based on similarity, those heatmaps further strengthen our approach of considering genes with peaks identified with at least 3 of the 4 antibodies as targets of SALL4-HDAC2.

Furthermore, if the rationale for employing two antibodies for each protein is improving specificity, then the authors should select only peaks identified by both antibodies. Otherwise, the authors should choose one of the two antibodies for downstream analyses, in particular if one of them presents a reliable/expected signal as has been reported in the literature with confidence. This is particularly striking in the case of HDAC2 antibodies: ab#1 identifies 3,500 peaks, ab#2 almost 10,000. Is it possible that ab#1 simply did not work? Checking publicly available HDAC2 ChIP-seq/CUT&RUN data would be helpful in this context.

Above, we have elaborated in much detail on our experimental choice of using different antibodies per factor and on our analytical choice of filtering the targets genes based on whether they show CUT&RUN sequencing peaks for at least 3 of the 4 antibodies used. Of note, we had used a 3rd antibody against HDAC2 (Abcam, ab7029) that was however eliminated from the analysis after bioinformatic analysis because it did not result in reliable peak callings. However, the other 4 antibodies used in this study showed reliable peaks after artifact peaks exclusion by negative controls, in silico trimming of repetitive regions and PCR/seq biases, so that we trust in the accuracy of those experiments.

Again, we believe that using two different antibodies per factor does not weaken our results, but oppositely, strongly strengthens our findings. As the reviewer noted correctly, the HDAC2 antibody #2 (Cell Signaling, 57156S) generated more than double the number of peaks as the antibody #1 (ab12169) did. However, the HDAC2 antibody #1 is actually well-described in literature and has been used for ChIP sequencing experiments (for instance DOI: [10.1155/2020/4384696](https://doi.org/10.1155/2020/4384696)), while antibody #2 has been less validated. Therefore, we do not fully agree that only considering the peaks of the HDAC2 antibody #2, whilst leaving away the peaks generated with antibody #1 is the most proper approach – even if more peaks were generated with antibody #2. Vice versa, and as elaborated on above, after applying stringent bioinformatic cut-offs, we feel that ignoring the peaks obtained for the HDAC2 antibody #2 (only because it is less cited in literature) would be an equally arbitrary, but possibly wrong approach. The same holds true for the two antibodies used for SALL4, which again was one of the rationales for our '3of4' approach.

All in all, we feel that focusing on peaks generated with at least 3 of 4 antibodies was the best compromise between on one hand a too stringent analysis (which would have been the analysis of peaks only shared between *all* antibodies) potentially ignoring many new and real peaks that are just detected by one antibody but not the other and on the other hand a less stringent analysis (which would have been to include all loci with at least one antibody peak per factor).

To clarify our approach for the reader, we now briefly elaborated on our choice of selecting genes with a at least peaks in 3 of 4 antibodies sampled in the revised manuscript (Lanes 230ff). Like that, our rational should be more evident for the reader and we thank the reviewer for having pointed out that issue.

Nevertheless, to address the reviewers concern and take up his/her suggestion of only focusing on the CUT&RUN peaks generated with antibodies reported in literature with confidence, we have re-analyzed the CUT&RUN peaks shared between the more common antibodies of each factor only (SALL4 Ab#1 (ab29112) and HDAC2 Ab#1 (ab12169)). We again correlated the newly obtained direct targets of only antibodies #1 with differential expression after SALL4 knock down and have ran Process Network enrichments on MetaCore™ as in Figures 4, 6 and 7. Below, to the reviewer's discretion, one can see that upregulated direct targets of SALL4 Ab#1 and HDAC2 Ab#1 still enrich in very similar pathways (such as *Cell adhesion*, *angiogenesis*, *TGFβ-signaling* and others, represented by *Integrins*, *TGFBR2*, *PDGFC* and others) as in our original analysis taking into account genes with shared peaks with at least 3 of 4 antibodies (Figure 6). Moreover, the downregulated direct targets of SALL4 Ab#1 and HDAC2 Ab#1 enrich in similar pathways (such as 'melanocyte development', represented by *MITF*, *DCT* and others) as in our original analysis (Supplementary Figure S15a, b).

Process Networks of **upregulated** targets of SALL4 Ab#1 (ab29112) and HDAC2 Ab#1 (ab12169)

Process Networks of **downregulated** targets of SALL4 Ab#1 (ab29112) and HDAC2 Ab#1 (ab12169)

Last but importantly, while we had already analyzed the SALL4 and HDAC2 DNA binding motifs considering both antibodies per factor, we have now for the revised manuscript also analyzed the motifs for the single antibodies. These novel motif analyses show similar binding motifs between the different antibodies for the same factor – 4 of the 5 top SALL4 motifs have the same best match and 5 of the top 8 HDAC2 motifs have the same best match - and further strengthen much the validity of all the antibodies that were used in our study. We have now also included the single antibody DNA binding motifs in the Supplementary Figures since this is useful information for the reader (Supplementary Figures S8, 9, 11, 12).

Also, the authors should adapt genomic snapshots in Fig 5e-i to be consistent with other figures in the paper, such as in Fig 7c-g (bar chart vs. line plot in IGV). Also, adding genomic coordinates along with a more comprehensive representation of gene TSS/exons/introns (as shown in IGV) instead of a stylized drawing, would be more appropriate here.

For consistency we have now changed the IGV tracks and plotted them all as bar charts in the revised figures as the reviewer suggested. Also, as further asked for, we have now added the intron/exon gene representation (as in IGV) to our selected gene tracks in Figures 5 and 7. However, we have decided to not further add the specific genomic coordinates (as an example: chr2:215,219,051-215,538,010) to each track because we feel it would not add any relevant information for the reader while making the figure unnecessarily overcrowded.

2. The integration of SALL4/HDAC2 CUT&RUN with RNA-seq and H3K27ac ChIP-seq is a key analysis in the paper, but is somehow missing. Although the authors have demonstrated that SALL4 is responsible for HDAC2-mediated downregulation of some invasive genes, the ChIP-seqs have not been integrated to put the whole story together.

We thank the reviewer for his/her careful revision of our revised manuscript and appreciate that he/her points out that a more thorough integration of the novel CUT&RUN data is needed.

While it is true that combining the SALL4/HDAC2 CUT&RUN and the acetylation ChIP-seq data is in principle a very good idea, in fact it is extremely difficult to bioinformatically choose a proper method for overlapping the two datasets. As a matter of fact, we do not necessarily expect a precise positional overlap between SALL4/HDAC2 and the loss of acetylation. This is due to the fact that deacetylation is the functional consequence of HDAC2 activity (conceivably recruited by SALL4), which could occur at any genomic locus that becomes functionally proximal to the SALL4/HDAC2 protein duet. This could also happen, for instance, via genomic looping of distant regulatory regions that come in proximity to SALL4/HDAC2. A similar problem in the field concerns the actual annotation of a peak to a gene, something that is performed based on proximity, but that is also known to be an imprecise approximation since an enhancer actually does not necessarily regulate its closest TSS.

Therefore, as it is very difficult to understand what are the genomic loci that the SALL4/HDAC2 complex regulate, we decided to look at the functional consequences of the SALL4-mediated HDAC2 activity, by discovering where loss of acetylation occurs genome-wide when SALL4 is downregulated. These loci are then compared with upregulated genes based on proximity to TSS (Figure 7i, j).

Nevertheless, to better integrate the novel SALL4 and HDAC2 CUT&RUN data with the ChIP-seq of H3K27ac after SALL4 knock-down, we show in a new density heat map (Figure 7h) that in the interval of 10 Kb where we have SALL4/HDAC2 peaks there are differential acetylation ChIP-seq peaks and that amongst those genes we find some invasiveness genes that are indeed upregulated after SALL4 knock-down such as TGFBR2, VEGFR-1 or ITGA6 (also listed in the novel Supplementary Data 9). This is an important new dataset, which shows "local" regulation of acetylation abundance in at least a fraction of the SALL4/HDAC2-bound loci. But consistently with the points mentioned above, it also shows that there is a large fraction of other SALL4/HDAC2 peaks that likely regulate acetylation and deacetylation at more distant regulatory regions.

In the revised manuscript, we have now added additional paragraphs on the novel Figure 7h and the issue discussed above (lanes 349ff).

Minor points:

- While the authors show some data in the Supp of Sall4 staining of human melanoma, it would be important to search publically available data sets, such as TCGA to see if SALL4 expression correlates with metastasis.

In fact, we did search TCGA for SALL4 expression and have not found a significant correlation with patient survival or disease stage (i.e. metastasis). Of note, this finding was somewhat to be expected since we hypothesize that SALL4 is a mediator of melanoma cell phenotype switching (the melanoma specific EMT), which while inducing cell invasiveness (upon SALL4 downmodulation), also leads to a reduction in cell proliferation. Hence the common idea is that while melanoma cells undergo a first switch towards increased invasion and reduced proliferation to allow dissemination from the primary tumor lesion and invasion into surrounding tissues and the circulation, a secondary switch - putatively due to secondary upregulation of SALL4 - reverting the cellular state back towards a highly proliferative (but less invasive) state is needed to establish fully blown secondary lesions and large-scale, detectable metastases. Therefore, we assume that the metastasis samples in data cohorts like TCGA present very advanced stages of melanoma in the sense that the peak of cellular invasiveness (intra- and extravasation from surrounding tissues and vessels) has already been passed and therefore such late-stage metastasis samples might not pose a good model to find key regulators of phenotype switching that are conceivably regulated in a very dynamic and quickly changing manner.

- The heatmap illustrating RNA-seq data in Fig 4a could be better represented by volcano plot highlighting significant genes.

We thank the reviewer for his/her input to the presentation of our RNA seq data. In fact, we have originally considered to present the significantly differentially changed genes as a volcano plot with the top changed genes highlighted. However, we preferred to show the obtained data in a more unbiased way as it stands now, where the reader can simply appreciate that there are comparable number of genes up- and downregulated after SALL4 (Figure 4a). Next, the pathway analysis (Figure 4b, c) shows the overall enrichment within biological processes. Of course, this data set was generated with genes ranked according to their degree of expression change after SALL4, however by including all significantly changed genes and not just a fraction of the top changed genes (as would be highlighted in a Volcano plot).

- The term RT-PCR used in the result section is misleading when referring to qRT-PCR, as correctly stated in the method paragraph. Please correct.

Thank you for having pointed this out; we have made the necessary changes in the revised manuscript.

REVIEWERS' COMMENTS

Reviewer #2 (Remarks to the Author):

Authors have addressed the concerns.

Addressing Reviewers Comments for Nat Commun from June 2021 (Revision III (editorial))

REVIEWER COMMENTS:

Reviewer #2 (Remarks to the Author):

Authors have addressed the concerns.

We thank reviewer #2 for his/her positive feedback and are happy that we have with this second re-revision satisfactorily addressed his/her points of concern.